# Biological, Equilibrium and Photochemical Signatures of C, N and S Isotopes in the Early Earth and Exoplanet Atmospheres

**DOI:** 10.3390/life15030398

**Published:** 2025-03-03

**Authors:** James R. Lyons

**Affiliations:** Planetary Science Institute, 1700 Fort Lowell, Tucson, AZ 85719, USA; jlyons@psi.edu

**Keywords:** biosignatures, atmosphere, exoplanets, early earth, isotopes

## Abstract

The unambiguous detection of biosignatures in exoplanet atmospheres is a primary objective for astrobiologists and exoplanet astronomers. The primary methodology is the observation of combinations of gases considered unlikely to coexist in an atmosphere or individual gases considered to be highly biogenic. Earth-like examples of the former include CH_4_ and O_3_, and the latter includes dimethyl sulfide (DMS). To improve the plausibility of the detection of life, I argue that the isotope ratios of key atmospheric species are needed. The C isotope ratios of CO_2_ and CH_4_ are especially valuable. On Earth, thermogenesis and volcanism result in a substantial difference in δ^13^C between atmospheric CH_4_ and CO_2_ of ~−25‰. This difference could have changed significantly, perhaps as large as −95‰ after the evolution of hydrogenotrophic methanogens. In contrast, nitrogen fixation by nitrogenase results in a relatively small difference in δ^15^N between N_2_ and NH_3_. Isotopic biosignatures on ancient Earth and rocky exoplanets likely coexist with much larger photochemical signatures. Extreme δ^15^N enrichment in HCN may be due to photochemical self-shielding in N_2_, a purely abiotic process. Spin-forbidden photolysis of CO_2_ produces CO with δ^13^C < −200‰, as has been observed in the Venus mesosphere. Self-shielding in SO_2_ may generate detectable ^34^S enrichment in SO in atmospheres similar to that of WASP-39b. Sufficiently precise isotope ratio measurements of these and related gases in terrestrial-type exoplanet atmospheres will require instruments with significantly higher spectral resolutions and light-collecting areas than those currently available.

## 1. Introduction

Stable isotope ratios are used as terrestrial biosignatures in all epochs of Earth history, from the present to the Archean, and are essential tools in astrobiology [1]. Most of these data are in situ measurements from ancient rocks. Here, my focus is on possible atmospheric isotope signatures. Models of Earth’s ancient atmosphere [2] carefully consider the evolution of atmospheric composition over time but generally do not consider the possible evolution of atmospheric isotope ratios. One of my objectives here is to estimate a plausible range of C isotope histories for the Earth’s early atmosphere. The geochemical record of C and N isotopes in ancient rocks will serve as a baseline to guide these estimates for early Earth and provide a possible framework for predicting isotopic biosignatures in the atmospheres of Earth-like exoplanets.

Several studies have been conducted on the detectability of isotope ratios in exoplanet atmospheres [3,4], including rocky planets in the Trappist-1 system [5]. Measurements have been reported for a young, directly imaged, super-Jupiter [6], and also for WASP-77b, which is a hot Jupiter [7]. In both cases, the C isotope ratio was determined for atmospheric CO, and these objects were found to be extremely enriched in ^13^C compared to Earth. Given the equilibrium temperatures of these two objects, they are not likely abodes for life, but their high ^13^C enrichment may suggest an accumulation of ^13^C-rich ices during formation beyond the CO snow line [6]. In addition to CO, Glidden et al. [8] have pointed out the favorability of using CO_2_ to determine C isotope ratios in exoplanet atmospheres.

Superimposed on any biological isotope fractionation in atmospheric molecules are photochemically induced isotope effects, with sometimes quite large isotope fractionation drive by UV photodissociation. Self-shielding is believed to be important in protoplanetary disks [9,10] and molecular cloud core [11,12] environments and is a likely mechanism for explaining the differences in C, N, and O isotope ratios for inner solar system planets compared to those of the Sun [11,13]. Self-shielding in N_2_ has been demonstrated to be important for ^15^N enrichment in nitriles in Titan’s atmosphere [14] and very likely occurred in Earth’s ancient atmosphere.

Photochemical self-shielding is an example of an abundance-dependent isotope fractionation process. Biological isotope fractionation is an example of a kinetic fractionation process driven by the mass dependence of relatively rapid biochemical reactions. In addition to these two mechanisms, equilibrium isotope fractionation is driven by differences in zero-point energies among isotopologues [15]. Equilibrium fractionation has its roots in the quantum mechanics of the harmonic oscillator for molecular bonds. These three isotope fractionation processes, equilibrium, kinetic, and abundance-dependent, encompass the majority of isotopic processes relevant to exoplanet atmospheres.

The remainder of this paper is organized as follows. I first discuss C and N isotope ratios in the modern Earth atmosphere. This is followed by model predictions for the equilibrium isotope ratios among pairs of C-containing and N-containing gases from [15]. Equilibrium isotope ratios are most relevant to warm and hot exoplanet atmospheres, but they serve as a useful benchmark for atmospheres with isotope ratios determined by photochemical or biological processes. Next, I examine how UV photochemical self-shielding in small molecules can dramatically alter C and especially N isotope ratios in photochemical products such as HCN. I also apply these ideas to S isotopes in warm Jupiters. I then address the evolution of C and N isotopes in Earth’s early atmosphere, allowing for terrestrial biological processes such as methanogenesis and nitrogen fixation. Finally, I present IR spectra of key atmospheric molecules and their isotopologues and briefly discuss the detectability of isotope ratios in exoplanet atmospheres with respect to the needed spectral resolution.

The objective of this paper is to consider the prospects of using stable isotope ratios as biosignatures in exoplanet atmospheres. The technical difficulty of astronomically observing isotope ratios is well appreciated [3,4,5,6,7], and it is unlikely that they will be useful as biosignatures on rocky worlds in the JWST era [16]. Nevertheless, isotopes offer a view of biological processes that is distinct from and complements the more usual atmospheric chemical composition arguments for biosignatures in atmospheres [17]. High-resolution spectral measurements with large telescopes will eventually make such measurements feasible, even for rocky planets.

## 2. C and N Isotope Ratios in the Modern Earth Atmosphere

As an example of how isotope ratios vary among molecules in the present-day atmosphere, I will first consider carbon in CO_2_ and CH_4_. Ignoring anthropogenic input, the CO_2_ mixing ratio is determined by the balance of CO_2_ emitted by volcanism and the sequestration of CO_2_ in marine carbonates, primarily by biological organisms. Rather than compute CO_2_ I will focus on the geochemcial processes that alter its isotope composition. The δ^13^C ratio of CO_2_ is determined primarily by CO_2_ exchange with the oceans. The massive reservoir of HCO_3_^−^ in the oceans and the high exchange flux between the atmosphere and oceanic mixed layer fix the C isotope ratios in the ocean. Neglecting CO_2_ photolysis in the upper atmosphere, a process that is not important for tropospheric isotope ratios, δ^13^C for CO_2_ is −8.4 permil today and was about −6.5 permil in the pre-industrial atmosphere [18]. The pre-industrial δ^13^C value for CO_2_ is close to the mantle value of −5 to −8‰ [19]. Photosynthesis, respiration, and air-sea exchange all contribute to the fractionation of C isotopes. Fractionation during photosynthesis creates fixed C with δ^13^C values 18 and 4‰ lower than those of atmospheric CO_2_ for C_3_ and C_4_ plants, respectively [20]. The predominance of C_3_ plants (about 85%) yields a terrestrial bulk biosphere with δ^13^C ~−22% relative to VPDB.

Expressing the δ^13^C of atmospheric CO_2_ in terms of its sources, we have(1)$$\delta^{13}C({CO}_{2})=\frac{\delta^{13}C(C_{plant})\varphi_{plant\;resp}+\delta^{13}C({CO}_{2,surf\;ocean})\varphi_{ocean\;exch}}{\varphi_{plant\;resp}+\varphi_{ocean\;exch}}$$
where *φ* is the CO_2_ flux due to either plant respiration or emissions associated with air-ocean exchange. The fluxes of C are *φ_plant resp_* = 60 Gt C yr^−1^ and *φ_ocean exch_* = 90 Gt C yr^−1^, and the representative δ-values are −22‰ for plants and +2.5‰ for surface dissolved inorganic carbon (DIC). Air-sea exchange results in kinetic and equilibrium isotope enrichment in HCO_3_^−^ relative to atmospheric CO_2_ of ~7–10‰ [21]. From Equation (1), the above values yield δ^13^C = −7.5‰ for the atmospheric CO_2_. Anthropogenic CO_2_ must also be included when computing δ^13^C values for the modern atmosphere [20].

To estimate the δ^13^C value for CH_4_ in the modern atmosphere, it is necessary to consider the production and loss processes. Methane is primarily a biogenic gas with a substantial anthropogenic contribution. The total source CH_4_ flux to the atmosphere is $\varphi_{{CH}_{4}}$ = 540 Tg CH_4_ yr^−1^, with 2/3 due to microbial activity (wetlands, rice paddies, animals, and termites) and 1/3 due to anthropogenic activity (biomass burning, landfills, coal mining, and natural gas) [22]. The sink flux of CH_4_ is almost entirely oxidation in the atmosphere via the reactionOH + CH_4_ → CH_3_ + H_2_O (2)
with a rate constant(3)$$k_{2}=1.36\times 10^{-13}{\left(\frac{T}{298}\right)}^{3.04}e^{-\frac{920}{T}}\;$$
The rate constant *k*_2_ has units of cm^3^ molec^−1^ s^−1^ and is valid from 195–1234 K [23].

Equating the sources and sinks yields(4)$$\varphi_{{CH}_{4}}=\int_{0}^{top}k_{1}[OH][{CH}_{4}]dz$$
OH and, to a lesser extent, CH_4_ have complicated vertical profiles, which I will approximate as following the barometric law. The vertical integral in Equation (4) may then be approximated as multiplication by the atmospheric scale height(5)$$H_{a}=\frac{kT}{mg}$$
where *k* is the Boltzmann constant, *T* is the atmospheric temperature, *m* is the mean molecular mass, and *g* is the acceleration of gravity. For the modern troposphere, *H_a_* = 8 km. Solving for the steady-state CH_4_ concentration yields(6)$$[CH_{4}]=\frac{\varphi_{{CH}_{4}}}{k_{2}[OH]H_{a}}$$
Most of the CH_4_ oxidation occurs in the troposphere. For a representative lower troposphere temperature of 280 K, $k_{2}=4.2\times 10^{-15}$ cm^3^ s^−1^. The CH_4_ source flux is 540 Tg CH_4_ yr^−1^ circa 1990 [22], which corresponds to $1.3\times 10^{11}$ molec cm^−2^ s^−1^. For a maximum OH number density in the troposphere of ~1 × 10^6^ cm^−3^, the CH_4_ number density is 3.9 × 10^13^ cm^−3,^ corresponding to a surface mole fraction of 1.5 ppm. This compares well with the 1991 atmospheric value of 1.7 ppm [24].

To compute the C isotope ratio for atmospheric CH_4_, I rewrite Equation (6) for ^13^C-specific processes. Most CH_4_ sources are either directly biogenic or involve the combustion of biogenic materials and will therefore produce a ^13^C-depleted CH_4_ flux compared to oceanic carbon. Additionally, oxidation reaction 2 has a rate constant that varies with the isotope. For the ^13^C isotopologue Equation (6) becomes(7)[CH413]=φCH413k213[OH]Ha
The global bulk source flux of CH_4_ has a δ^13^C of ~−53‰ [25]. This value is the weighted average of biogenic sources (wetlands, rice fields, and ruminants) with δ^13^C ~−70 to −50‰ and thermogenic/pyrogenic sources (biomass burning) with δ^13^C ~15‰ [25]. There is a kinetic isotope effect in reaction 2 that causes slower oxidation of ^13^CH_4_ by about 4‰ at 296 K [26]. The resulting CH_4_ isotopologue ratio is(8)CH413CH4=φCH413φCH4k2k213

From the definition of the geochemical δ-value for the C flux, δ^13^C, the ratio of fluxes may be expressed as(9)φCH413φCH4=δ13C(φsource)+1C13C12VPDB
In Equation (9), δ^13^C has a fractional rather than permil value, δ^13^C(*φ*) + 1 = 0.9470, and VPDB is the Vienna Peedee Belemnite isotope standard with ^13^C/^12^C = 0.011180 [27]. For k2k213 = 1.0039 [26], Equations (8) and (9) yield δ^13^C(CH_4_) = −49‰, which is in good agreement with the measured tropospheric value of −47‰ [28]. There is additional isotope fractionation in the stratosphere due to the oxidation of CH_4_ by O(^1^D) and Cl [29,30], which I will neglect here. Aerobic methane oxidation by methanotrophs also produces a very large positive ^13^C enrichment in unoxidized CH_4_; however, this does not significantly affect the δ^13^C of the large reservoir of CH_4_ in the modern atmosphere [31].

In both ancient and modern Earth atmospheres, the primary nitrogen species is N_2_. In the modern atmosphere, the next most abundant N-containing compounds are nitrogen oxides (N_2_O, NO, NO_2_, and HNO_3_), whereas in the ancient Earth atmosphere, HCN and possibly NH_3_ were important species. For Archean CH_4_ ~1000 ppm, photochemical models predict HCN concentrations of ~100 ppm [2]. NH_3_ is rapidly lost by photolysis in early Earth atmosphere models unless a haze (probably hydrocarbon) is present to block near-UV radiation. In the modern Earth atmosphere, N isotope variability is predicted and observed in nitrogen oxide species due to photolysis and isotope exchange reactions. Rather than discussing this in detail here, I refer the reader to a paper by Michalski et al. [32]. Both HCN and NH_3_ are present in the atmosphere today, but at mixing ratios of ~200 ppt [33] and ~1–10 ppb [34], respectively. Nitrogen isotope measurements of NH_3_ indicate two primary sources, agriculture and livestock and fossil fuel-related activities. Agriculture and livestock produce NH_3_ with δ^15^N ~−40 to −10‰ and fossil fuel sources produce NH_3_ with δ^15^N ~−20 to +10‰ [34]. Marine values for NH_3_ are ~+5 to +13‰. All δ^15^N values are reported relative to an atmospheric N_2_ standard.

## 3. Equilibrium C, N and S Isotope Ratios

Isotopic equilibrium is an important end-member for the isotope ratios in planetary atmospheres. For an equilibrium reaction involving isotope exchange among gas-phase species, the equilibrium constant can be defined in terms of the partition functions of the reactants and products. Following Richet et al. [15], I write an isotope equilibrium reaction as(10)$$AX_{n}+BX_{m}^{\prime }=AX_{n}^{\prime }+BX_{m}$$
where X’ is an isotope of X, and n and m are integers. A fractionation factor α is defined as(11)$$\alpha (AX_{n},BX_{m})=\frac{R(AX_{n})}{R(BX_{m})}$$
where the ratio *R* is(12)$$R(AX_{n})=\frac{\left[AX_{n}^{\prime }\right]}{\left[AX_{n}\right]}$$

For relatively small delta-values (<100‰), the fractionation factor and δ’s are related by(13)$$1000ln\alpha \left(AX_{n},BX_{m}\right)=\delta (AX_{n})-\delta (BX_{m})$$
where $\delta (AX_{n})=10^{3}\left(R_{X}(AX_{n})/R_{X}(std)-1\right)$. For n = m = 1, the fractionation factor becomes the equilibrium constant for the reaction. For more general cases, the fractionation factor may be expressed as [15](14)$$\alpha (AX_{n},BX_{m})=K{\left(AX_{n},BX_{m}\right)}^{\frac{1}{mn}}\frac{\varepsilon (AX_{n})}{\varepsilon (BX_{m})}$$
where ε is an ‘excess factor’ and K is the equilibrium constant for the reaction $mAX_{n}+nBX_{m}^{\prime }=mAX_{n}^{\prime }+nBX_{m}$. The formulation of the equilibrium constants and excess factors is in terms of the partition functions that are a function of the vibrational and rotational energy levels available to the primary and rare isotopologues, as determined from the Schrodinger equation. Richet et al. [15] give a more detailed description of the excess factor ε.

To tabulate results it is convenient to define an additional fractionation factor β, which is the fractionation factor between AX_n_ and X and may be written as [15](15)$$\beta (AX_{n},X)=\frac{{\left[\frac{X^{\prime }}{X}\right]}_{AX_{n}}}{\left[\frac{X^{\prime }}{X}\right]}=\frac{R(AX_{n})}{R(X)}$$
The two fractionation factors are then related by(16)$$\alpha (AX_{n},BX_{m})=\frac{\beta (AX_{n},X)}{\beta (BX_{m},X)}$$
In terms of partition functions, β is given by(17)$$\beta (AX_{n},X)={\left[\frac{Q(AX)}{Q(AX_{n})}\right]}^{\frac{1}{n}}{\left[\frac{m^{\prime }}{m}\right]}^{-\frac{3}{2}}\varepsilon (AX_{n})$$
where m and m’ are the masses of the isotopes X and X’. The partition function Q has its usual form(18)$$Q=Q_{tr}Q_{e}Q_{vib}Q_{anh}Q_{rot}Q_{rot-vib}$$
which accounts for the population of translational, electronic, vibrational, and rotational states, including the effects of anharmonicity and rotational-vibrational coupling. The vibrational partition function is of particular importance because it accounts for most of the temperature dependence of equilibrium isotope fractionation.

The computed β-factors for C exchange as a function of temperature for CO, CO_2_, CH_4,_ and HCN are shown in Figure 1a, and the β-factors for N exchange for N_2_, NH_3,_ and HCN are shown in Figure 1b. The difference in δ^13^C values, Δδ^13^C, for several pairs of molecules illustrates the expected range of C isotope ratios under conditions of isotopic chemical equilibrium (Figure 2a). At temperatures above 700 °C, the magnitude of Δδ^13^C < 15‰ and decreases with increasing temperature. For hot Jupiters, where CO and to a lesser extent, CO_2_ [35] are the primary carbon-bearing molecules, Δδ^13^C is likely to be too small to detect. Isotope-selective photodissociation of CO (i.e., self-shielding) and/or CO_2_ will produce non-equilibrium isotope ratios. For hot Jupiters, these effects are primarily in the upper atmosphere. CO self-shielding will produce ^13^C-depleted CO and ^13^C-enriched C atoms. If the ^13^C-rich C atoms are sequestered in a C-rich haze layer, a detectable depletion of ^13^C in CO may be detectable. If the ^13^C-rich C is ionized to C^+^, rapid isotope exchange with CO will occur, erasing the self-shielding signature. Isotope fractionation due to CO_2_ photolysis will occur at longer wavelengths and greater depths in the atmosphere. At greater depths in the atmosphere, C isotope exchange between CO and CO_2_ will occur as a result of reactions with H and OH that efficiently inter-convert CO and CO_2._

Cooler, rocky exoplanets (T < 500 K) are likely to have N_2_ atmospheres with CO_2_ or CH_4_ as the primary C-bearing molecules [36]. For temperatures below 400 °C, the magnitude of Δδ^13^C > 20‰ relative to CO_2_ and increases with decreasing temperature. This well-known equilibrium isotope effect arises from the preference of the heavier isotopes for the higher bond strength compound. For CH_4_ in the most common habitable temperature range of 0 to ~50°C, Δδ^13^C ~−60 to −80‰ relative to CO_2_ at equilibrium. For gases to reach isotopic equilibrium at low temperatures, a catalyst must be present. Life may play the role of a catalyst, although biochemical reactions are generally dominated by kinetic isotope effects.

Nitrogen isotopes do not show as large a range of equilibrium isotope fractionation as the C isotopes (Figure 2b). At hot Jupiter temperatures, Δδ^15^N of only a few permil is predicted for HCN and NH_3_ relative to N_2_. Even at habitable temperatures, where N_2_ is plausibly the primary atmospheric gas (for an Earth-sized planet), Δδ^15^N for HCN relative to NH_3_ is only ~2‰. These are both astronomically observable molecules, but remotely measuring such a small difference in δ-values is difficult due to the large uncertainties expected for isotope ratio measurements. A small Δδ^15^N is consistent with the magnitude of the fractionation that occurs during N_2_ fixation by nitrogenase. Measurements by [37] of δ^15^N of biomass produced by diazotrophic growth of several bacterial strains yielded ~−1 to −3‰ relative to air N_2_ for traditional bacteria using the most common Mo-Fe nitrogenase. Bacteria using either V-Fe or pure Fe nitrogenase show larger fractionation of −3 to −7‰, but this is still quite small compared to biogenic C isotope fractionation.

Sulfur isotopes exhibit equilibrium fractionation intermediate between the C and N isotopes (Figure 2c). Motivated by the photochemical models of Tsai et al. [37] for WASP-39b, SO_2_, H_2_S, and S_2_ are considered. Other high-temperature S-species include OCS and CS. Of possible relevance to cooler rocky planets, SO_3_ and CS_2_ δ^34^S values are also computed. I have not considered biological fractionation of S isotopes. With an atmospheric temperature of ~900 °C, WASP-39b will have a Δδ^34^S ~3‰ for SO_2_ relative to H_2_S. However, photochemical self-shielding is likely to substantially modify these values, as discussed in the following section.

## 4. Photochemical Signatures of C, N and S Isotopes

Any assessment of biosignatures in isotope ratios must consider the possibility of photochemically generated signatures. Photochemistry drives atmospheric chemistry away from the thermodynamic equilibrium. If the UV photon flux, atmospheric temperature, and pressure are sufficiently stable, a steady-state photochemical composition will be reached. I consider here how photochemistry can alter the C, N, and S isotope ratios in planetary atmospheres. This is most applicable to terrestrial-type exoplanets, but may have relevance to warm Neptunes and the lower temperature range of hot Jupiters. It is not my intent to exhaustively cover this topic but rather to consider a few specific cases in a qualitative or semi-quantitative manner.

For high-temperature exoplanets, C and N will be present in the IR-detectable region of the atmosphere as primarily CO and N_2_. These two diatomic molecules are isoelectronic with the ground states of X^1^Σ^+^ and X^1^Σ_g_^+^, respectively. Both molecules undergo photodissociation in the far-ultraviolet (FUV) region, from 91 to 108 nm for CO and from 91 to 100 nm for N_2_ molecules. They are also photodissociated and photoionized at wavelengths <91 nm; however, for H_2_-rich environments, the onset of H ionization at 91 nm absorbs most FUV photons, as would be expected for warm Neptunes and hot Jupiters. Both CO and N_2_ undergo predissociation, which means that the absorption of a sufficiently energetic FUV photon creates a bound electronically excited state that then either relaxes by fluorescence or crosses to an unbound dissociating state, resulting in atomic products. Again, for both CO and N_2,_ the bound excited states have relatively long lifetimes (e.g., ~10 psec to 1 nsec), resulting in narrow transition line widths (~0.5 to 0.005 cm^−1^). Isotope substitution in these molecules, for example, ^13^CO and ^29^N_2_, yields an absorption spectrum very similar to the main isotopologues (^12^CO and ^28^N_2_) but offset by several to several 10’s of cm^−1^. This means that the photodissociation of a column of either CO or N_2_ will result in a process termed self-shielding, in which the most abundant isotopologue (^12^CO or ^28^N_2_) will become optically thick and cease to dissociate, while the rare isotopologue (^13^CO or ^29^N_2_) will continue to undergo photodissociation. The result is a massive enrichment of the rare isotopes of the dissociation products, i.e., ^13^C, ^15^N, or ^17^O and ^18^O when considering self-shielding by N_2_ or CO, together with a depletion of the rare isotopes in the parent molecules. These isotope effects occur downstream of the region where the primary isotopologues become optically thick in the FUV. Self-shielding is a specific case of isotope-selective photodissociation in which isotope substitution induces changes in the absorption spectrum of a molecule due to changes in the coupling of electronic states. There are numerous examples of molecular clouds, protoplanetary disks [9,38,39], and planetary atmospheres [14,40].

I first consider N isotope fractionation due to N_2_ self-shielding in the early Earth atmosphere. Photodissociation of N_2_ and CH_4_ likely produces substantial quantities of HCN [41,42], and the self-shielding of N_2_ could produce a large ^15^N enrichment in HCN. As demonstrated by Oro [43], HCN is an essential ingredient in the prebiotic formation of adenine. To assess this possibility, I use model results from [42] for the early Earth (0–100 km) combined with an estimate of the downward flux of ^28^N_2_ and ^29^N_2_ from a model of Earth’s upper atmosphere (120–1000 km) (Lyons and Bondoc, in prep). Self-shielding of N_2_ occurs primarily at altitudes of 150–300 km. Rather than use the high-resolution cross sections for N_2_, I use shielding functions as described in [38] for CO. Assuming a 45° solar zenith angle and a pure N_2_ atmosphere, shielding functions have been determined for self-shielding by ^28^N_2_ and ^29^N_2_, and for mutual shielding by ^28^N_2_ and ^29^N_2_ (on ^29^N_2_ and ^28^N_2_, respectively) [39]. The photodissociation rate coefficients for ^28^N_2_ and ^29^N_2_ are written as(19)J28(z)=J0Θss(NN228)Θms(NN229)(20)J29(z)=J0Θss(NN229)Θms(NN228)
where Θ*_ss_* and Θ*_ms_* are the self-shielding and mutual-shielding functions, *N* is the column density of ^28^N_2_ or ^29^N_2_ from the top of the atmosphere to altitude *z*, and *J*_0_ is the N_2_ dissociation rate at the top of the atmosphere. Equations (19) and (20) only account for self-shielding from 91 to 100 nm. Although N_2_ dissociates down to 80 nm, line broadening makes the self-shielding effect less significant. For the modern atmosphere, *J*_0_ ~2 × 10^−7^ s^−1^, and for the early Earth atmosphere, it is ~10^3^ times higher [42]. The shielding functions for N_2_ are not given here (they are lengthy polynomial expressions) but will be described elsewhere. The photodissociation rate coefficients for ^28^N_2_ and ^29^N_2_ illustrate the self-shielding effect (Figure 3). At a given altitude between 350 and 120 km, *J*_29_ > *J*_28_ due to self-shielding by ^28^N_2_. The shielding functions in Equations (19) and (20) capture the effects of line saturation in ^28^N_2_ without having to integrate the high-resolution cross sections over wavelength. The decrease in the photolysis rate coefficient for ^28^N_2_ is nearly a factor of 10 at maximal self-shielding altitudes (Figure 3), yielding ^15^N enrichment in N atoms of ~1000 permil. In addition to a much higher solar EUV flux, the solar wind particle flux was also much higher, with implications for atmospheric erosion for small, weakly magnetized planets such as Mars [44], a scenario I will not consider here.

Line broadening will diminish the self-shielding effect. For planetary atmospheres both Doppler and pressure broadening need to be considered. The Doppler linewidth is given by(21)$$\Delta \nu_{D}=\nu_{0}\frac{v_{th}}{c}$$
For the modern Earth thermosphere, the thermospheric temperature is ~1000 K, and the thermal gas velocity (most probable speed) is *v*_th_ = 7.7 × 10^4^ cm s^−1^. At a wavelength of 100 nm, the Doppler linewidth is Δν_D_ = 0.26 cm^−1^. This linewidth is much smaller than the typical spacing between the rotational lines for N_2_ and CO and does not impact self-shielding. Pressure broadening derives from the decoherence time due to molecular collisions, and is given by(22)$$\Delta \nu_{p}\sim \frac{1}{4\pi }\sigma_{c}{nv}_{th}$$
where σ_c_ ~3 × 10^−15^ cm^2^ is the molecular collision cross-section, and *n* is the atmospheric number density. At low pressures (<1 microbar), where self-shielding occurs for N_2_ in Earth’s atmosphere (Figure 3), Δν_p_ ~10^−8^ cm^−1,^ which is negligible.

Photolysis of N_2_ produces ground state and excited state atoms in approximately an equal mixture as(23)N2+hν→N(S4)+N(D2)
N(^2^D) decays to N(^4^S) with a radiative lifetime of 6.1 × 10^4^ s (17 h). It undergoes quenching to N(^4^S) by collisions with N_2_ and CO on timescales of ~60 s at 100 km in the model from [42]. N(^2^D) can also react with species such as H_2_ via the reaction(24)N(D2)+H2→NH+H
The model of Tian et al. [42] has an H_2_ mixing ratio of 2 × 10^−3^ at 100 km, which implies a loss timescale for N(^2^D) by reaction 22 of ~1400 s for a thermosphere temperature of 180 K. All of these timescales are much shorter than the model eddy transport timescale at 100 km of ~3 × 10^5^ s. The recombination of N atoms via $N+NH\to N_{2}+H$ acts to reverse the effects of N_2_ self-shielding.

A maximum downward flux of ^14^N and ^15^N at altitude *z* can be estimated by ignoring N loss due to N + NH recombination, which yields(25)φ14=2∫ztopJ28N228dz(26)φ15=∫ztopJ29N229dz
For the modern Earth upper atmosphere model used here, *φ*_14_ = 7.45 × 10^8^ cm^−2^ s^−1^ and *φ*_15_ = 1.76 × 10^7^ cm^−2^ s^−1^ at 120 km. Zahnle [41] estimates a download N flux of ~1 × 10^10^ cm^−2^ s^−1^ for wavelengths from 79.6 to 91.2 nm. Relative to the modern Earth atmosphere N_2_ with ^15^N/^14^N = 1/272.0, the N atom downward flux due to 91–100 nm photons has δ^15^N(*φ_N_*) = 5400‰. Making the end-member assumption of no self-shielding for 80–91 nm photons, the overall (80–100 nm) δ^15^N(*φ_N_*) = 370‰. The true value will be between 370 and 5400‰.

Exchange reactions can also be very important and may include the reactions(27)N15+N228↔N15+N229(28)N15(D2)+N228↔N14(D2)+N228
For both reactions, the intermediate N_3_ is a doublet, and therefore, reaction 27 is spin-forbidden. From the work of [45], I infer a rate constant of $k_{27}<2.6\;{\times \;10}^{-13}e^{-\frac{12,600}{T}}$, which makes reaction 27 negligibly slow at temperatures < 1000 K. If this is not the case, and reaction 27 is important, it will greatly reduce the δ^15^N enrichment of N atoms (Figure 4). Reaction 28 is spin-allowed but does not appear to have a measured rate coefficient. At 100 km, the timescale for the loss of ^15^N(^2^D) by reaction 28 is comparable to the timescale for loss by quenching by N_2_ for *k*_28_ ~2 × 10^−14^ cm^−3^ s^−1^, a plausible value for a spin-allowed reaction. Further work on reaction 28 is needed, as it too can reduce the N atom δ^15^N enrichment (Figure 5).

HCN is produced by two pathways [41,42](29)N+CH23→HCN+H(30)$$N+CH_{3}\to H_{2}CN+H\overset{H}{\to }HCN+H_{2}+H$$
Both of these pathways, which peak at ~70 km in the Tian et al. model [42], will impart a large ^15^N enrichment to HCN, specifically δ^15^N(HCN) ~370–5400‰ from the results for *φ_N_* above. N_2_ self-shielding occurs at considerably higher altitudes (~120–350 km) than does HCN formation. The loss of HCN occurs primarily by photolysis at Ly α (121.6 nm) to make products H + CN. The UV absorption cross sections for HCN show significant structure both near Ly α and in a vibrational progression from 130–150 nm (Figure 6). The photolysis of HCN is likely to modify the isotope ratios in HCN, as suggested by the large vibrational peak shifts between HCN and DCN. If HCN is optically thick, its ^15^N enrichment from formation would be reduced. In summary, a significant isotopic photosignature of HCN is a strong possibility for terrestrial-type exoplanets. Because this signature would be enriched in ^15^N, it should be distinct from any biogenic HCN in the atmosphere. Finally, I note that there is some evidence for ^15^N enriched Archean organic sediments (e.g., +15‰, ref. [46]), but these data are from metamorphosed sediments that likely preferentially lost ^14^N during burial and heating.

Carbon isotopes are also affected by photolysis reactions. The spin-forbidden photolysis of CO_2_(31)CO2+hν(>167nm)→CO+O(P3)
was predicted [47] and experimentally demonstrated [48] to produce CO with a large depletion of ^13^C. The large fractionation arises from a small redshift in the absorption spectrum at longer wavelengths for the ^13^CO_2_ isotopologue. The shift occurs in a region of the spectrum that is rapidly decreasing in absorption strength with wavelength, which enhances the effects of the spectral shift. This is also a region of the spectrum populated by hot bands (bands with a lower level vibrational quantum number *v* > 0), which makes the isotope fractionation strongly temperature-dependent. There is evidence for this process occurring in the present-day atmosphere of Mars from occultation measurements of CO by the ExoMars spacecraft at Mars, which show that CO has a δ^13^C of ~−250 to −150‰ [49,50], consistent with CO_2_ photolysis as the primary source of CO gas in the Martian atmosphere and consistent with photochemical models [40].

In Earth’s atmosphere today, CO has a concentration of ~100 ppb and is produced from biomass burning, CH_4_ oxidation, and as a byproduct of anthropogenic combustion reactions. ACE-FTS solar occultation measurements of CO in the Earth’s mesosphere reveal isotopically depleted CO at altitudes of ~50–80 km with δ^13^C ~−160‰ [51]. This isotope signature exhibits seasonal and transport-related variations in the mesosphere and upper stratosphere but not in the lower stratosphere and troposphere.

Venus provides another example of a very low δ^13^C value for CO in the upper atmosphere. Observations of the *J =* 1 → 2 transition for ^12^CO and ^13^CO at 230 and 220 GHz in the 80–110 km region of the atmosphere revealed ^12^CO/^13^CO = 185 ± 69 [52]. Converting this ratio to δ-values relative to the PDB standard yields δ^13^C(CO) = ${-520}_{-130}^{+290}\;$‰. The high end of this range (−230‰) is comparable to those observed on Earth and Mars. I calculated the photodissociation rate coefficients for ^12^CO_2_ and ^13^CO_2_ using equations analogous to Equations (19) and (20) and the ab initio cross sections of [47] (rather than shielding functions) together with CO_2_ number densities from [53]. Although the measurement uncertainties are large, the ^13^C depletion in CO is certainly due, at least in part, to reaction 31 (Figure 7). Ueno et al. [48] suggest that the cross sections in [47] overestimate C isotope fractionation by 30%, suggesting that the computed δ^13^C values may be too small in magnitude to account for the observations. CO is optically thick up to ~110 km in the Venus mesosphere; therefore, it is possible that ^12^CO self-shielding at wavelengths <108 nm contributes to the ^13^C depletion in CO. CO_2_ would shield CO at these wavelengths; therefore, CO self-shielding should be diminished. Venus provides an excellent example of a photochemical-derived isotope signature that we can expect in other rocky planets with CO_2_ atmospheres.

Of more relevance here is the early Earth atmosphere with a high CO_2_ partial pressure and a correspondingly high CO abundance. For the early Earth, ref. [41] predicts a CO mixing ratio of 2 × 10^−5^ at the ground and 1 × 10^−3^ at 100 km. The primary source of CO is the photolysis of CO_2_ by reaction 31 and at higher altitudes by spin-allowed photolysis of CO_2_ at wavelengths <167 nm (which does not cause large isotope fractionation). As CO_2_ photolysis is the primary source of CO, a large negative δ^13^C signature is expected to be present in the middle atmosphere. The reaction $CO+OH\to CO_{2}+H$, a primary loss pathway for CO, will act to decrease the ^13^C enrichment in CO. The net result is a low δ^13^C value (~−100 to −200‰) for CO in the middle atmosphere, which may eventually be detectable in analogous rocky exoplanets using cross-correlation techniques.

I next briefly consider C isotope fractionation due to CO self-shielding in the early Earth atmosphere and in an H_2_-rich exoplanet atmosphere. CO will be optically thick and, therefore, undergo self-shielding at a lower altitude than N_2_ because of its lower abundance. Because both molecules have long-lived predissociation states, the mutual shielding of N_2_ on CO will occur but will not significantly diminish the self-shielding in CO. I therefore expect large ^13^C and ^17^O and ^18^O enrichments in product C and O atoms. The fates of C and O atoms differ. C is likely to react with OH to reform CO via C + OH → CO + H, which would act to reverse the effects of CO self-shielding. If an organic haze is present, ^13^C-enriched C could be sequestered in the haze, leaving measurable ^13^C-depleted CO in the atmosphere. Isotopically enriched O atoms are likely to eventually form H_2_O, although the initiating reaction, O + H_2_ → OH + H, is strongly temperature-dependent. Exchange reactions between O and O_2_ and OH and H_2_O will diminish the self-shielding signature in O isotopes, but a positive δ^17^O and δ^18^O in atmospheric H_2_O is possible, although it is likely to be diluted by H_2_O already in the atmosphere.

I next consider CO self-shielding in a warm Neptune/hot Jupiter. At very high temperatures, isotopically enriched O will rapidly form OH and H_2_O, and the reaction OH + CO → CO_2_ + H and the reverse reaction will act to erase the self-shielding signature in O-containing species. In addition, the exchange reaction(32)Ox+CO16↔O16+COx
which has an activation energy of 6.9 kcal mole^−1^ (activation temperature of 3470 K), will also reduce the self-shielding signature. The fate of the ^13^C-enriched C atoms is less clear. C may react with H_2_ to form CH_2_ in the 3-body reaction C+H2→MCH23 where M is a 3rd-body molecule (H_2_ most likely). The rate constant is 6.9 × 10^−32^ cm^6^ s^−1^ at 300 K [54], and is likely somewhat slower at higher temperatures. This reaction is too slow at CO self-shielding altitudes, which are likely to be ~microbars of total pressure. Radiative recombination, C+H2→CH2+hν3, is likely a faster pathway with a typical rate constant of ~1 × 10^−12^ cm^3^ s^−1^, and thus a loss timescale for C of ~1 s. The formation of ^3^CH_2_ would lead to CO reformation via reactions such as CH2+O→CO+H2 or CO+2H3, or more indirectly by CH2+O→CH+OH3, followed by the reaction of CH with O to form CO. Reformation of CO by any of these reactions will decrease the magnitude of the self-shielding signature in C. Another possible mechanism for erasing a CO self-shielding signature is the exchange of C with CO via(33)C13+CO12↔C12+CO13
This reaction should proceed through a C_2_O intermediate with a triplet ground state, which is spin-allowed. I did not find a 2-body exchange rate coefficient for reaction 33; however, there is a measured 3-body rate coefficient for C_2_O formation of 6.31 × 10^−32^ cm^6^ s^−1^ [55].

If C produced from CO photolysis under self-shielding conditions avoids recombination to CO, one possible fate for it would be to form a haze layer, either by contributing to an existing organic haze layer or by forming a solid carbon haze layer at high altitudes via the reaction C→C(s). Depending on the temperature, C particles can condense as either amorphous carbon or graphite. The haze particles would be ^13^C-enriched, and the CO would be ^13^C-depleted at altitudes in the vicinity of the CO self-shielding. The latter may be detectable at infrared wavelengths, as a low ^13^C/^12^C ratio.

Finally, I briefly consider the sulfur isotopes in hot Jupiters. The detection of photochemically produced SO_2_ in the hot Jupiter WASP-39b [37] provides the prospect of measuring the exoplanet ^32^S/^34^S isotope ratio modified by UV self-shielding. Self-shielding in SO_2_ has been investigated as a mechanism for explaining Archean S isotope signatures in sedimentary rocks [56,57]. SO_2_ undergoes predissociation in the C-X band from about 180–220 nm and has an absorption spectrum with resolved lines in a vibronic progression. Sulfur isotope substitution creates a redshift in this spectrum, which generates a large isotope enrichment of the less abundant stable isotopes (^34^S, ^33^S, and ^36^S) in the dissociation product SO. Although this mechanism does not explain the early Earth rock record, it has been verified experimentally [58]. The key requirement is an optically thick column of SO_2_.

For WASP-39b, the peak mixing ratio for SO_2_ is ~5 × 10^−5^ at ~0.1 mbar on the morning terminator and ~2 × 10^−5^ at ~0.03 mbar on the evening terminator [37]. For a scale height of 890 km, these correspond to vertical column densities of 2.9 × 10^18^ cm^−2^ and 3.9 × 10^17^ cm^−2^ in the morning and evening, respectively. The peak cross-section for SO_2_ near 200 nm is ~3 × 10^−17^ cm^2^ at room temperature and ~2 × 10^−17^ cm^2^ at ~1000 K. (It should be noted that the absorption lines are narrow; therefore, accurate UV cross sections require high-resolution laboratory measurements). Peak vertical optical depths are ~58 and 7.8, morning and evening. The tangential optical depths along the terminators are ~25 times the vertical optical depths. Thus, at the peak SO_2_ mixing ratios, SO_2_ is strongly optically thick under all conditions. Self-shielding by ^32^SO_2_ will produce a large enrichment in ^34^SO during photolysis(34)SOx2+hν→SOx+O
where *x* = 32 or 34 (and also 33 and 36, but these are much less abundant). Self-shielding by ^32^SO_2_ means that *J*_34_ >> *J*_32_, so the instantaneous number density of ^34^SO will be highly enriched compared to the standard isotope ratios. (The sulfur isotope standard is a meteorite mineral, the Canyon Diablo troilite or CDT). Models [56] and measurements [57,58] suggest δ^34^S ~100 to 200‰ enrichments in SO. The steady-state enrichment of ^34^SO and the corresponding depletion of ^34^SO_2_ depend on subsequent and concurrent reactions. SO and SO_2_ are produced by oxidation reactions with OH, which is produced by the photolysis of H_2_O [37]. Although photolysis is the primary loss process for SO_2_ in WASP-39b, the continual reformation of SO and SO_2_ will act to reduce the self-shielding signature. Additionally, S exchange with SO,(35)SOx+S↔SO+Sx
may also reduce the self-shielding enrichment of ^34^SO unless SO photolysis is the primary source of S atoms. Detailed photochemical modeling is required to address these issues.

## 5. Evolution of C and N Isotopes in Earth’s Early Atmosphere

From the discussion above, it is clear that significant isotope fractionation is possible in exoplanet atmospheres. Is it possible to see biosignatures against this backdrop of equilibrium and photo-induced isotope fractionation, and what might this have looked like in early Earth? Catling and Zahnle [2] produced an evolution diagram of the composition of Earth’s atmosphere over the past 4 billion years (Figure 8). I have added HCN to this diagram by scaling HCN relative to CH_4_ before and after the atmospheric Great Oxidation Event (GOE): 10^−1^$P_{{CH}_{4}}$ prior to the GOE and 10^−4^$P_{{CH}_{4}}$ after the GOE. More accurate HCN evolution calculations can be performed. Prior to the GOE, the primary loss process for HCN is photolysis; after the GOE, loss by OH dominates, although the rate of this loss reaction will vary depending on the O_3_ partial pressure.

Catling and Zahnle [2] assumed a gradual onset of methanogenesis at ~4.0 Gyr and a gradual onset of photosynthesis at ~3.0 Gyr. Thus, in their model, the high CH_4_ partial pressure (2000 ppm) at 4.0 Gyrs may be mostly biogenic. Prior to the onset of methanogenesis, but after the short-lived massive impact-generated reduced atmospheres, CH_4_ is primarily thermogenic and/or derived from geothermal and hydrothermal reactions. Thermogenic refers to CH_4_ produced by the high-temperature decomposition of buried organic compounds. Today and during the Phanerozoic, most buried organics are biogenic in origin; therefore, the δ^13^C values for atmospheric CH_4_ are relatively low at ~−20 to −50‰ (Figure 9) [59]. Prior to 4.0 Gyrs, most buried organics were either derived from the original delivery of chondritic materials or from the burial of material produced by the UV processing of impact-reduced atmospheres. Bulk insoluble organic matter (IOM) in carbonaceous chondrites has δ^13^C ~−35 to −10‰ and in ordinary chondrites has δ^13^C ~−24 to −10‰ [60], coincidentally similar to the range for C_3_ and C_4_ plants. It is therefore likely that thermogenic CH_4_ produced from 4.4 to 4.0 Gyr would have a similar range of δ^13^C values to what we see today (Figure 9).

Geothermal and hydrothermal CH_4_ differ from thermogenic CH_4_ in that the former is derived from inorganic reactions in the Earth’s crust. The general form of the reaction responsible for the geothermal/hydrothermal production of CH_4_, known as the Sabatier reaction, is $CO_{2}+4H_{2}↔CH_{4}+2H_{2}O$ [61]. Serpentinization, which generates H_2_ in the crust via a redox reaction between Fe(II) and H_2_O, also falls into this category of reactions. I will not assess the possible flux of CH_4_ prior to methanogenesis, but I will assume that the flux of CH_4_ available from crustal sources was sufficient to maintain a substantial CH_4_ partial pressure.

Following Catling and Zahnle [2], I assume that methanogenesis becomes the dominant CH_4_ source by ~4.0 Gyr. If hydrogenotrophic methanogenesis is the primary initial methanogenic process (Figure 9), then substantial ^13^C depletion is likely to be present in atmospheric CH_4_ (Figure 10). If acetoclastic methanogenesis was the primary source of CH_4_, then atmospheric CH_4_ would have been less strongly fractionated. Carbon isotope analysis of organics in ancient rocks yields a mean of δ^13^C ~−30‰ with a large variation [62] (Figure 10). If hydrogenotrophic methanogenesis was the predominant source of CH_4_ in the ancient atmosphere, then the organics in rocks have δ^13^C values decoupled from atmospheric CH_4_. Anaerobic methane oxidation by methanotrophs, which is known to produce extremely light δ^13^C in lipids (<−100‰) and leave correpsondingly ^13^C-enriched CH_4_, could also modify the CH_4_ δ^13^C history, as shown in Figure 10. In the modern atmosphere, anaerobic methanotrophy does not significantly alter δ^13^C for atmospheric CH_4_ (see Section 2). For the early Earth, it has been argued that anaerobic methanotrophs can explain Tumbiana Formation organics with δ^13^C ~−60‰ at 2.8 Gyr ago [63] (not shown in Figure 10). This event could have produced atmospheric CH_4_ with δ^13^C values of >>−50‰. Given the above considerations, the evolution of methanogenesis and δ^13^C for atmospheric CH_4_ in Figure 10 must be treated as an approximation of the actual values.

The reservoir of dissolved inorganic carbon (DIC) in the oceans today is ~38,000 Pg C, vastly larger than the sum of atmosphere, land plants, and soil C, which is ~2800 Pg C [22]. Even for a slightly more acidic early ocean, C would be present primarily as HCO_3_^−^ and CO_3_^2−^, and I assume that δ^13^C of early DIC is approximately the same as modern DIC. I make this assumption because air-sea exchange is the primary source of equilibrium fractionation between atmospheric CO_2_ and DIC [21]. Photosynthesis in surface waters creates additional ^13^C enrichment in DIC and yields δ^13^C(DIC) ~+1 to + 3‰ [64]. The majority of carbonates in sedimentary rocks are believed to be biogenic, and the δ^13^C of carbonates has generally been fairly close to 0‰ over the past 3.5 Gyrs, with occasional deviations of ±10‰ [65]. I will further assume that abiotic carbonate formation on the prebiotic Earth did not strongly affect the C isotope ratios in atmospheric CO_2_, while CO_2_ mostly tracked marine carbonate formation over the past 3.5 Gyrs with a roughly constant fractionation due to air-sea exchange. Figure 10 illustrates the evolution of δ^13^C for CO_2_ over the past 4.4 Gyrs, with the constant value of −6.5‰ [64] prior to 3.5 Gyr ago supplanted by the curve (thick gray line) that tracks the carbonates. The CO curve represents the minimum δ^13^C expected due to CO_2_ photolysis. This minimum occurs at a high altitude in the mesosphere or even the lower thermosphere for a high *p*CO_2_ case. In the lower atmosphere, the δ^13^C of CO is closer to that of CO_2_.

**Figure 10 life-15-00398-f010:**
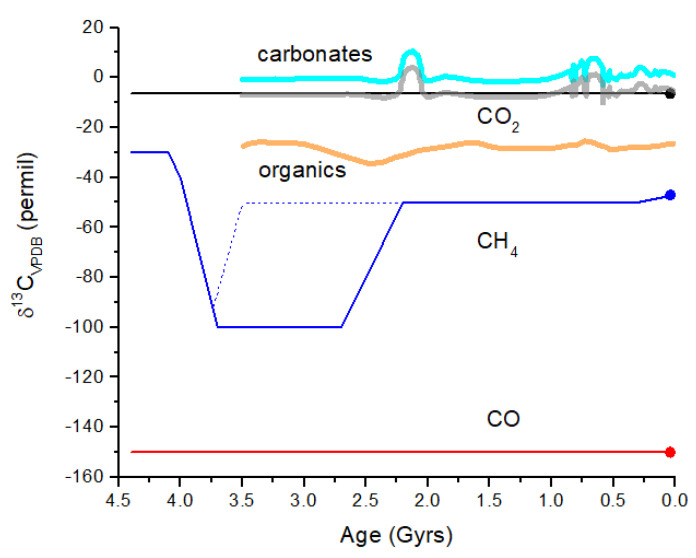
Schematic representation of δ^13^C evolution in the Earth’s atmosphere. In the simplest scenario, CO_2_ is derived from volcanism and has δ^13^C ~−6.5‰ as for mantle C. CO_2_ undergoes gas exchange with the surface ocean throughout Earth history since 4.4 Gyr ago (black line). More likely, CO_2_ tracked the carbonates. The carbonates curve (thick cyan line) is a smoothed version of measurements of δ^13^C for marine carbonates over the past 3.5 Gyr [65]. The curve illustrates the approximate range of variation seen in marine carbonates, which would likely be present in atmospheric CO_2_. Atmospheric CO_2_ will track carbonates but will be displaced by about −6.5‰ (thick gray curve). The CO curve illustrates the effect of isotope fractionation during CO_2_ photolysis in the upper atmosphere. Although δ^13^C for CO is constant over time, the CO mixing ratio decreases by many orders of magnitude from ~10^−3^ at 4 Gyr to ~10^−10^ today. The CH_4_ curve (blue line) illustrates abiogenic sources prior to 4.0 Gyr ago, followed by hydrogenotrophic methanogenesis for the next ~1.5 Gyr, and then CH_4_ with modern δ^13^C values after the GOE at 2.4 Gyr ago. The mean δ^13^C for organics recorded from rocks (thick orange line) [62] does not exhibit highly negative δ^13^C, indicating that either the rock record does not entirely correlate with atmospheric CH_4_ or that methanogenesis was not a major biochemical process (dotted blue line). The δ^13^C difference between CO_2_ and abiogenic CH_4_ is ~25‰ versus ~95‰ between CO_2_ and hydrogenotrophic methanogen CH_4_. This comparison of CO_2_ and CH_4_ may represent a viable biosignature in a given terrestrial exoplanet atmosphere.

From Figure 10, the maximum difference in δ^13^C for atmospheric CH_4_ and CO_2_ is Δδ^13^C = δ^13^C(CH_4_) − δ^13^C(CO_2_) ~−95‰ during an era when hydrogenotrophic methanogenesis was dominant. In the modern atmosphere, this difference is about −40‰, and in the prebiotic atmosphere, this difference may have been ~−25 to −30‰. For Earth-like exoplanets (i.e., rocky planets with oceans), the value of Δδ^13^C may serve as a biosignature, assuming similar microbial evolution. Δδ^13^C is an entirely internal measure, meaning that a comparison with the δ^13^C of the parent star or other objects in the planetary system is not necessary. The feasibility of measuring Δδ^13^C in an exoplanet atmosphere is considered in the following section.

## 6. Spectral Signatures of Isotopic Gases

Here, I briefly consider the HITRAN line intensities [66] for the relevant isotopologues discussed above to illustrate the magnitude of the isotope-induced band shifts. A complete analysis of the detectability of the isotope ratios discussed above should include isotopologue cross sections, line broadening, and detection in a noisy spectrum [4,67], which will be discussed elsewhere.

The ν_2_ and ν_3_ bands of CO_2_ show redshifts of 19 cm^−1^ and 70 cm^−1^, respectively, for the ^13^CO_2_ isotopologue (Figure 11). These large redshifts reflect the large displacement of the C atom in the bending and asymmetric stretch modes of CO_2_. As discussed in [16], the large band shifts in CO_2_ make isotopologue ratio measurements possible in exoplanet atmospheres with large atmospheric scale heights using JWST MIRI or NIRSpec at medium spectral resolution (*R* ~1000–3000). The CO molecule [66] also exhibits large isotopic band shifts of ~50 cm^−1^ and ~95 cm^−1^ for the fundamental (4.7 microns) and first overtone (2.35 microns). CO is especially useful for measuring isotope ratios in hot Jupiters [6] and brown dwarfs [7], and is essential for determining the C isotope ratio of the solar photosphere [13]. In addition, CO is the molecule of choice for measuring the C and O isotope ratios in exoplanet parent stars.

**Figure 11 life-15-00398-f011:**
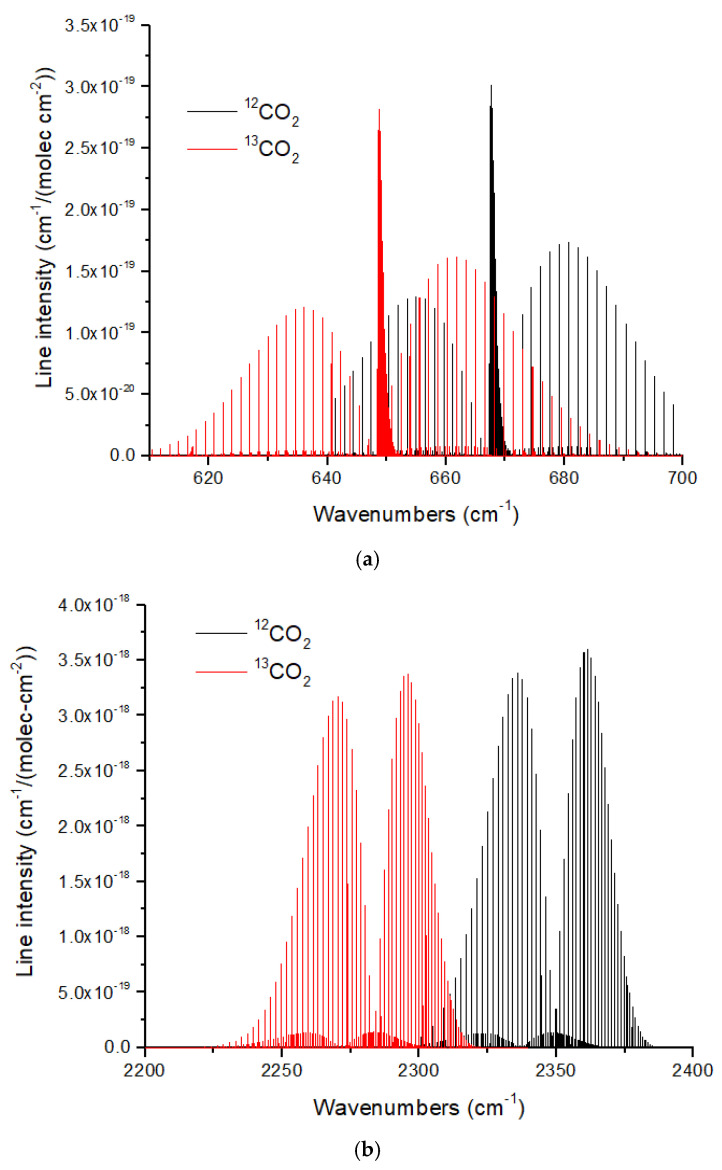
(**a**) Normalized line intensities at 296 K for the ν_2_ band (bending mode) of ^12^CO_2_ and ^13^CO_2_. The redshift of the ^13^CO_2_ spectrum is ~19 cm^−1^. (**b**) For the CO_2_ ν_3_ band (asymmetric stretch) the redshift is larger at ~70 cm^−1^. For both figures, the ^13^CO_2_ line intensity has been normalized by the ^13^C fraction (0.011057) given in HITRAN. Data were obtained from HITRANonline [66].

The ν_3_ (3.3 microns) and ν_4_ (7.7 microns) bands of CH_4_ also show redshifts upon ^13^C substitution. However, these redshifts are ~8–10 cm^−1^ (Figure 12), which are smaller than those for CO and CO_2_. In addition, there is an occasional overlap in the ν_4_ band. This means that some ^13^CH_4_ lines will be masked by lines from the ~100 times more abundant ^12^CH_4_. This problem is more acute for the ν_3_ band, for which there is a coincidental overlap of the P and R branch lines adjacent to the Q-branch, such that pressure broadening will cause increased line overlap and masking of ^13^CH_4_. The Q branches of both bands are a complex cluster of lines with a typical line spacing of ~0.05–0.10 cm^−1^. Resolving these lines would require a high spectral resolution of *R* ~30,000–60,000, far beyond the JWST capability. Resolving the Q-branch envelopes is easier, requiring a minimum *R* ~160 and 380 for ν_4_ and ν_3,_ respectively.

As discussed above, HCN photochemically produced in a N_2_-CO_2_-CH_4_ atmosphere, as proposed for the Archean Earth, is expected to have a massive ^15^N enrichment (δ^15^N ~300 to several 1000‰) due to N_2_ self-shielding at wavelengths <100 nm. HCN is a linear molecule with a C-H stretch mode (ν_1_), a doubly degenerate bending mode (ν_2_), and a C-N stretch mode (ν_3_). Only the ν_3_ mode has substantial N displacement, but it is a very weak band. The much stronger ν_1_ and ν_2_ bands have substantial C displacement but only slight N displacement (Figure 13a). The result is a very small ν_2_ band shift of ~1–1.5 cm^−1^ for HC^15^N, causing overlap between the Q branches (Figure 13b). To resolve these Q branches requires a minimum resolution of *R* ~700, and to resolve individual lines requires *R* ~3500–7000. Hydrogen isocyanide, HNC, with a central N atom, will exhibit larger band shifts for H^15^NC, although it is generally less abundant than HCN.

As a final spectroscopic example, I consider ^32^SO_2_ and ^34^SO_2_. The SO_2_ ν_3_ band at 1360 cm^−1^ is strong and exhibits a band shift of 18 cm^−1^ (Figure 14a). The spectrum is densely populated with transitions, which means that the line overlap from ^32^SO_2_ may shield some of the ^34^SO_2_ lines. Line overlap is an important consideration when retrieving the isotope ratios. Significant line overlap by the more abundant isotopologue will create an artificially low abundance for the rare isotopes in this case, ^34^SO_2_. This could give the impression of a stronger self-shielding signature in SO_2_ than is actually present. The typical spacing between the lines can be assessed from Figure 14b. The line spacing for a given isotopologue for ν_3_ is ~0.03 to 0.2 cm^−1^, which requires a spectral resolution of *R* ~6700–45,000.

Model calculations by Tsai et al. [37] suggest that SO may also be observable in WASP-39b or similar atmospheres. HITRAN does not have line intensity data for ^34^SO, but estimating the fundamental vibrational frequency from the ^32^SO fundamental gives(36)$$\nu_{34}=\nu_{32}\sqrt{\frac{\mu_{32}}{\mu_{34}}}$$
where μ is the reduced mass of each isotopologue. For ν_32_ ~1125 cm^−1^, ν_34_ ~1114 cm^−1^ for a red shift of 11 cm^−1^. Reaction 34 will result in a significant enhancement of ^34^SO, which, together with the less dense SO spectrum, suggests that it may be easier to obtain a S isotope ratio from SO than from SO_2_.

## 7. Discussion

The study of isotopes in exoplanet atmospheres is still in its early stages. The few measurements made thus far have been in hot Jupiter (or brown dwarf) atmospheres. These have yielded surprising results for C isotope ratios, with evidence for extreme enrichment of ^13^CO, interpreted as the input of a high fraction of ^13^C-enriched ices during planet formation [6]. Confirmation of high ^18^O enrichment would support this interpretation. It is clear that exoplanet isotope ratio data can contribute to understanding the formation environment of exoplanets, just as they do in understanding the formation of our solar system [9,11]. The measurement of the C and O isotope ratios of CO in parent stars would be a valuable contribution to this effort. A careful assessment of line overlap and dipole moment functions is essential, as was found for the solar photosphere [13,68].

A recent review of isotope ratios as biosignatures in exoplanet atmospheres concluded that the prospects are bleak, given the current technology [16]. Obtaining quality isotope ratios for cooler terrestrial-type exoplanets is not possible with the JWST (insufficient resolution) or ground-based telescopes (insufficient number of photons at high resolution). I entirely agree with Glidden et al.’s [16] assessment of rocky planets. However, technology and analysis methods will improve, and it is important to establish the range of fractionation possible for various isotope systems. This includes both photochemically produced isotope signatures and potential biologically generated isotope signatures.

Glidden et al. [16] argue that an evaluation of exoplanet isotope biosignatures requires a context such as the isotope ratios of the parent star or the parent molecular cloud. Our own solar system argues against this suggestion. The solar photosphere is 400‰ depleted in ^15^N compared to the Earth’s atmosphere [69], yet this appears to have minimal implications with respect to N_2_ fixation, the key nitrogen biochemical process on Earth.

Table 1 presents a summary of the isotopic results obtained here for several types of exoplanet atmospheres, from rocky planets to hot Jupiters. Exo-Venus objects imply CO_2_ rich atmospheres with depleted H_2_O, while exo-Earths are similar to a pre-oxygenated early Earth. Oxygenated exo-Earths would likely have lower mixing ratios of CH_4_ and HCN, making it more difficult to measure the isotope ratios of these molecules. The category of super-Earths covers a very broad range of compositions from essentially Earth-like at ~300 K to much higher temperatures with a retained, primary atmosphere. I will use the term super-Earth for a strictly rocky object, and for a super-Earth-sized object that has retained its primary atmosphere, I will use the term sub-Neptune [70]. The range of atmospheric compositions possible for super-Earths/sub-Neptunes is vast [71]. Here, I will treat rocky super-Earths as exo-Earths in terms of atmospheric composition, with the recognition that many super-Earths have equilibrium temperatures far above that expected to be capable of supporting life. Sub-Neptunes and warm Neptunes are H_2_-rich objects with substantial H_2_O and possibly CH_4_, NH_3,_ or N_2_. If NH_3_ is the primary form of nitrogen, then HCN will not have a significant ^15^N enrichment. Photochemical hazes, including organic hazes, will diminish the magnitude of photochemically-derived isotope signatures. CO and CO_2_ in the presence of substantial OH from H_2_O photolysis will have diminished C isotope signatures due to interconversion mediated by OH. HCN produced photochemically from N_2_ will have a reduced δ^15^N signature at higher temperatures due to weaker self-shielding and a lower mixing ratio of CH_4_. For hot Jupiters, SO_2_ self-shielding presents the interesting possibility of ^34^S depletion in SO_2_ and large ^34^S enrichment in SO. At shorter wavelengths, CO self-shielding could produce a detectable δ^13^C signature in CO if the photochemical product C is sequestered in a C-rich haze. I have not included the photochemical signature of CO_2_ due to CO_2_ photolysis in Table 1. It might have a detectable ^13^C enrichment, but this will generally be much smaller in magnitude than that of CO for rocky exoplanets.

The next steps in the analysis presented here are twofold. First, the infrared cross sections for the isotopologues discussed are computed with a complete treatment of line broadening for relevant atmospheric conditions. Second, to evaluate the ability of cross-correlation with absorption templates, with and without isotopes, to detect isotope ratios in simulated exoplanet transit data. Simulated data must include appropriate observational noise [68].

## 8. Conclusions

For photochemically-derived isotope signatures, far-UV self-shielding produces some of the largest isotopic enrichments known outside of nuclear reactions. Some of the most important molecules with respect to self-shielding are N_2_, CO, and SO_2_, all of which have either been detected in hot Jupiter atmospheres or are expected to be present. N_2_ self-shielding will yield ^15^N-enriched N atoms that can be sequestered in HCN or its tautomer HNC. In early Earth-like N_2_-CO_2_-CH_4_ atmospheres, HCN formation is likely to be efficient, and we can expect significant ^15^N enrichment (δ^15^N ~370 to 5400‰) in HCN, especially in the prebiotic atmosphere. Such high enrichments increase the likelihood of detecting the rare isotopologue. More detailed photochemical modeling is needed to evaluate the production of highly ^15^N-enriched HCN or HNC in warm Neptune or hot Jupiter atmospheres. Spectroscopic measurement of the ^15^N/^14^N ratio is complicated by the small band shift (~1.5 cm^−1^) of the ν_2_ band of HC^15^N. Hydrogen isocyanide, H^15^NC, will have a larger red shift in the ν_2_ band. The measurement of δ^15^N in these molecules for Earth-like exoplanets awaits future instrumentation.

A photochemically-derived signature in C isotopes is especially likely to be found in CO produced from CO_2_ spin-forbidden photodissociation. For a CO_2_ dominated atmosphere, highly depleted ^13^CO will be present in the mesosphere or lower thermosphere, as is well demonstrated by CO in the mesosphere (80–110 km) of Venus, which I argue is a result of reaction 31. Venus can serve as a model exoplanet for photochemical C isotope signatures.

Sulfur isotopes present an interesting scenario because of the presence of photochemically produced SO_2_ in the hot Jupiter WASP-39b. The morning and evening terminator column densities for SO_2_ are optically thick in the C-X dissociation band around 200 nm. SO_2_ self-shielding will produce SO that is highly enriched in ^34^S. Either ^34^S-enriched SO or possibly ^34^S-depleted SO_2_ may be observable in hot Jupiter atmospheres with the JWST. For the ν_3_ band of SO_2_, line overlap among isotopologues must be carefully evaluated due to the high density of rovibrational lines.

Isotopic biosignatures in general exhibit much smaller isotope fractionation than does photochemical self-shielding, but this fractionation is of far greater significance. I argue here that what matters most are the isotopic differences in pairs of atmospheric molecules involved in key biochemical processes. Important pairs include CO_2_/CH_4_ and N_2_/NH_3_. Isotope equilibrium argues that a significant difference in isotope ratios is expected for CO_2_ and CH_4_ at low (~300 K) temperatures, but not for N_2_ and NH_3._. On prebiotic Earth, a typical difference in δ^13^C for CH_4_ and CO_2_ is ~−25‰. After the advent of hydrogenotrophic methanogenesis, the difference could be as large as Δδ^13^C = δ^13^C(CH_4_) − δ^13^C(CO_2_) ~−95‰. Living systems are not in chemical equilibrium; however, life processes may act to bring isotope ratios closer to isotopic equilibrium in low-temperature biological systems. Alternatively, biochemical kinetic isotope effects could produce isotope fractionation similar to equilibrium isotope fractionation at low temperatures.

This δ^13^C difference between atmospheric CH_4_ and CO_2_ is the isotopic biosignature that we seek. It is an internal measure independent of the δ^13^C of the parent star or other objects in the planetary system. It assumes a planetary surface that carries out photosynthesis, methanogenesis, and probably N_2_ fixation. The technology to measure isotope ratios with an accuracy of ~50‰ or better on an Earth-like exoplanet does not presently exist. However, it will eventually exist, and this internal comparison of isotope ratios will be a valuable biosignature.

## Figures and Tables

**Figure 1 life-15-00398-f001:**
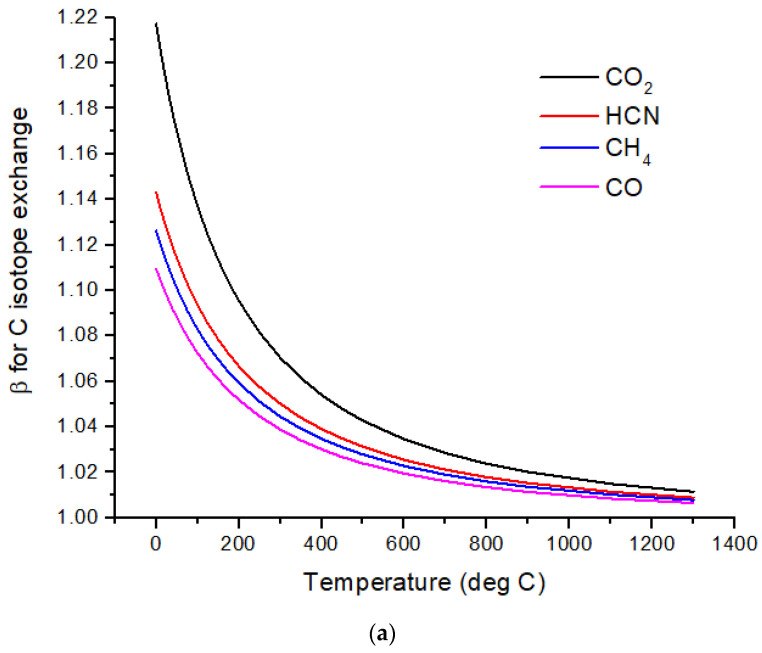
(**a**) Beta factors for C isotope exchange for several C-containing molecules relevant to exoplanet and early Earth atmospheres. (**b**) Beta factors for N isotope exchange for several N-containing molecules relevant to exoplanet and early Earth atmospheres. (**c**) Beta factors for ^34^S–^32^S isotope exchange for several S-containing molecules relevant to exoplanet and early Earth atmospheres.

**Figure 2 life-15-00398-f002:**
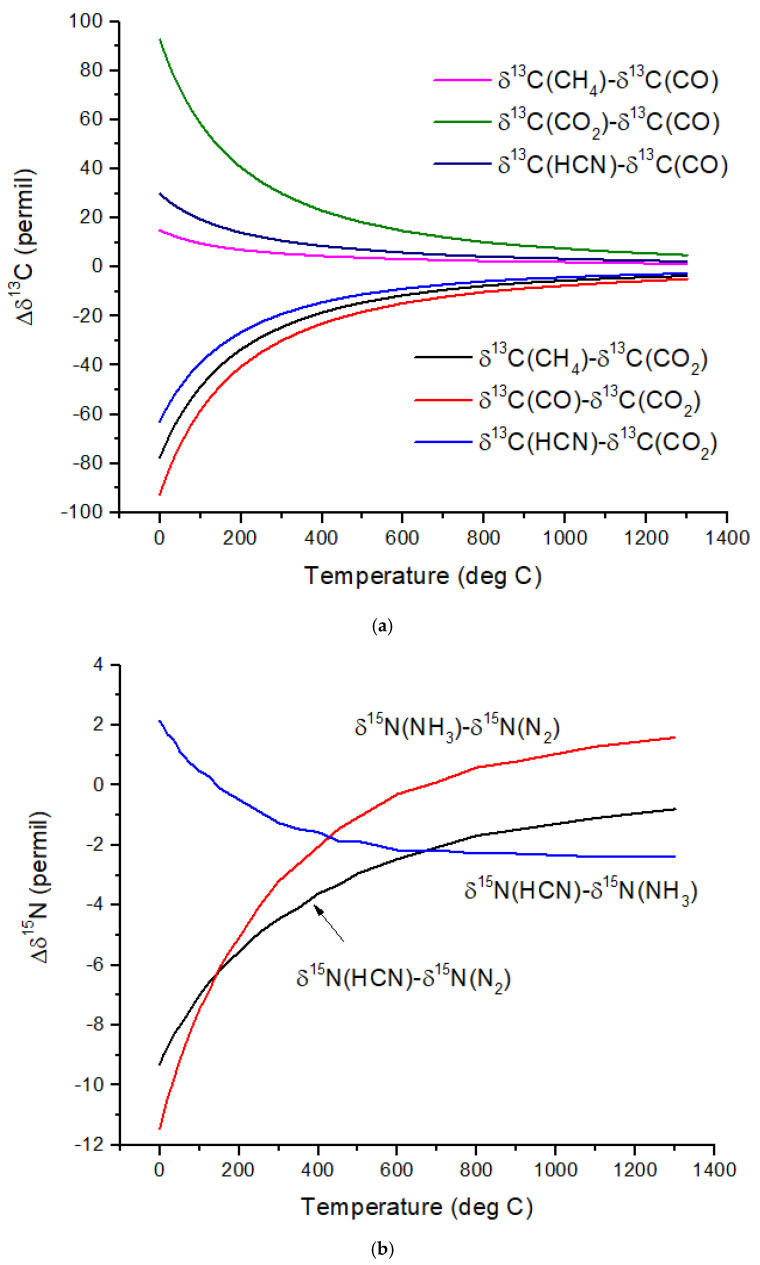
(**a**) Differences in equilibrium δ^13^C values, defined as Δδ^13^C, for several pairs of C-containing molecules relevant to exoplanet atmospheres. Attaining equilibrium at lower temperatures would require the presence of a catalyst. (**b**) Differences in equilibrium δ^15^N values, defined as Δδ^15^N, for several pairs of N-containing molecules relevant to exoplanet atmospheres. N_2_ is included here as an important reference, but its isotope ratio cannot be determined from infrared observations. (**c**) Differences in equilibrium δ^34^S values, defined as Δδ^34^S, for several pairs of S-containing molecules relative to H_2_S and relevant to exoplanet atmospheres. S_2_ is included here as an important reference, but its isotope ratio cannot be determined from the infrared observations.

**Figure 3 life-15-00398-f003:**
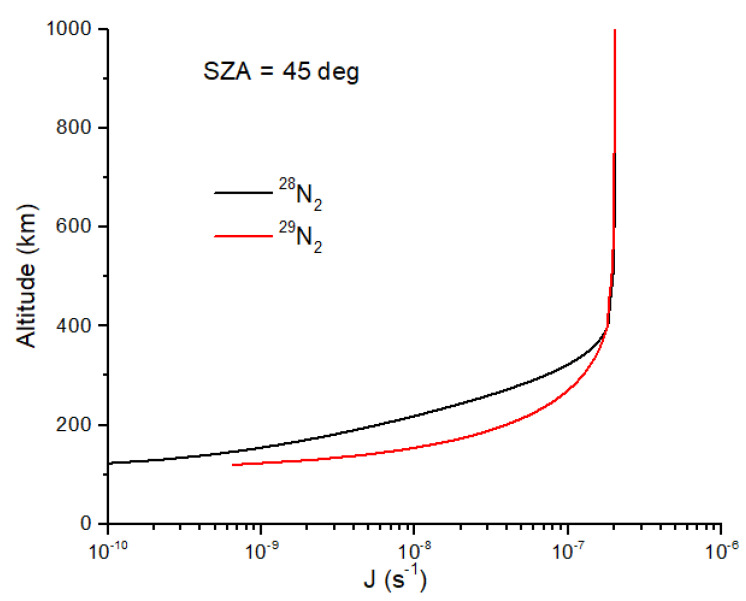
Photodissociation rate coefficients for ^28^N_2_ and ^29^N_2_ isotopologues for an Earth atmosphere model for a solar zenith angle (SZA) of 45°. The more abundant ^28^N_2_ is optically thicker at a given altitude than is ^29^N_2_. This defines the self-shielding region to be from ~350 km to 120 km, but it should be noted that the rate coefficient has decreased by two orders of magnitude by 120 km.

**Figure 4 life-15-00398-f004:**
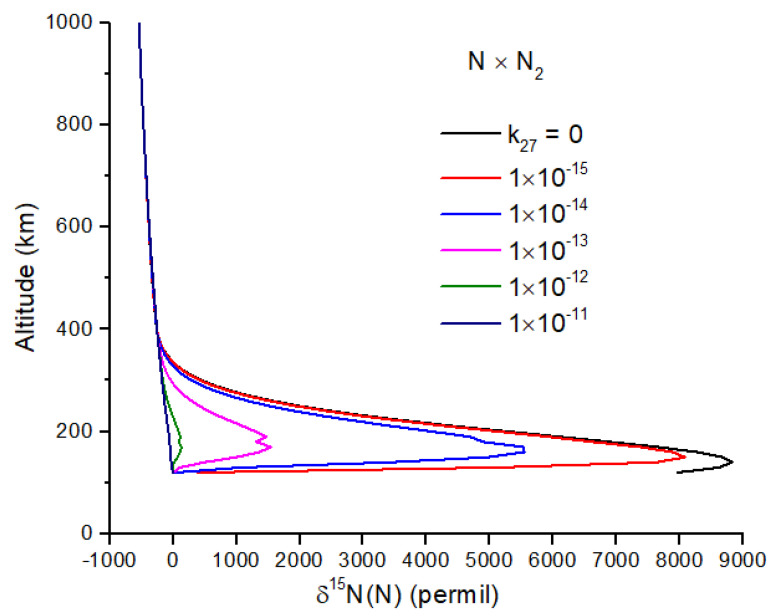
δ^15^N of N atoms due to N_2_ self-shielding for several values of the rate constant for N exchange with N_2_. Measurements at 1273 K indicate that this exchange reaction is effectively zero at lower temperatures [45], implying that the black curve is the most plausible. Peak enrichment occurs from 150–200 km. The possible recombination of N with NH is not included here.

**Figure 5 life-15-00398-f005:**
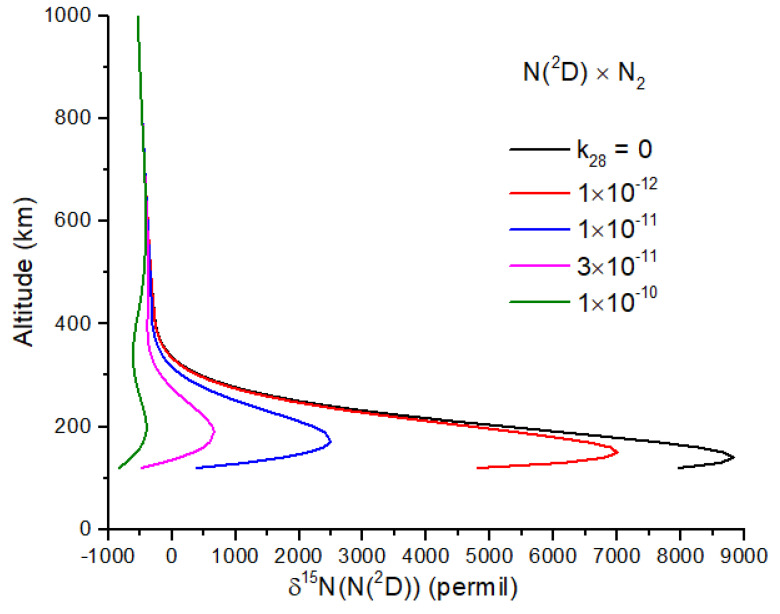
δ^15^N of N(^2^D) for several values of the rate constant for N(^2^D) exchange with N_2_. To the best of the author’s knowledge, this rate constant has not been measured.

**Figure 6 life-15-00398-f006:**
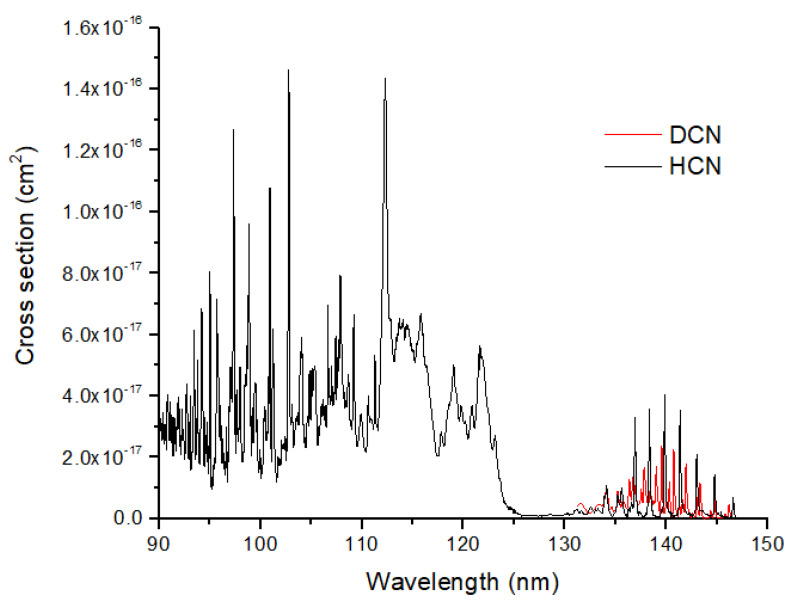
Measured UV cross sections of HCN and DCN.

**Figure 7 life-15-00398-f007:**
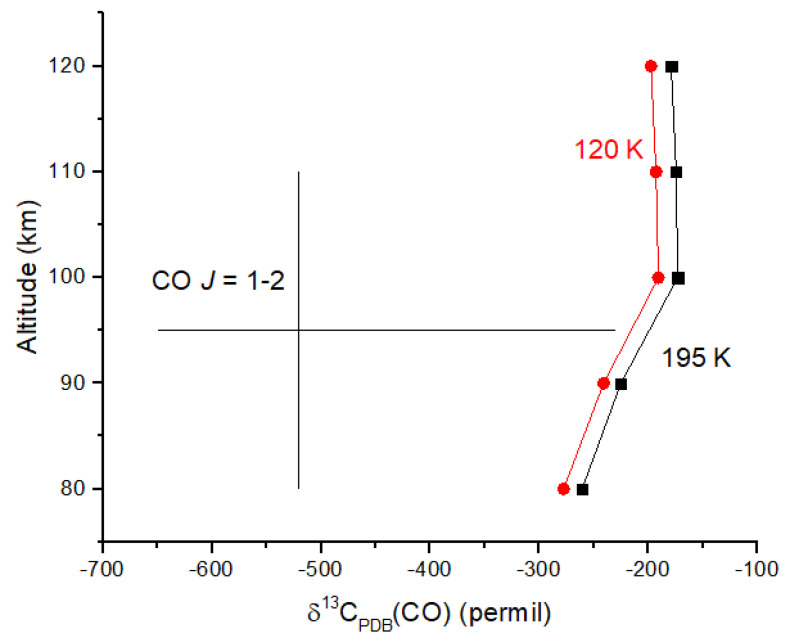
Computed carbon isotope fractionation due to CO_2_ photolysis in the Venus mesosphere using the computed isotopic cross sections at 195 K (black) and 120 K (red) [47]. Millimeter wave observations of ^12^CO/^13^CO from [52] have been converted to δ-values with uncertainties. Additional fractionation processes may be required.

**Figure 8 life-15-00398-f008:**
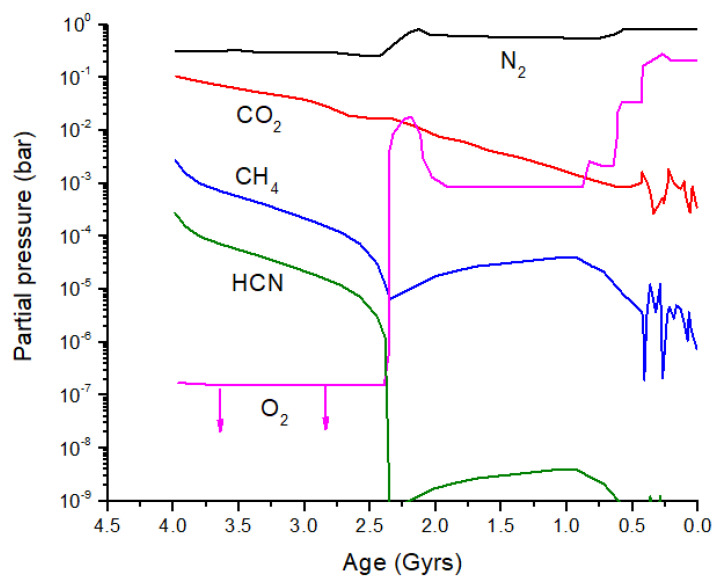
Composition of the Earth atmosphere over time. These are representative partial pressure curves with large uncertainties for ages >1 Gyr and very large uncertainties beyond 3 Gyr. The N_2_, CO_2_, CH_4,_ and O_2_ curves are from a review by Catling and Zahnle (2020) [2]. HCN is estimated to be 10^−1^ times the CH_4_ partial pressure prior to the Great Oxidation Event (GOE) at 2.4 Gyr and 10^−4^ times CH_4_ after the GOE. My focus here is on the detectable species CO_2_, CH_4,_ and HCN for ages > 2.5 Gyr.

**Figure 9 life-15-00398-f009:**
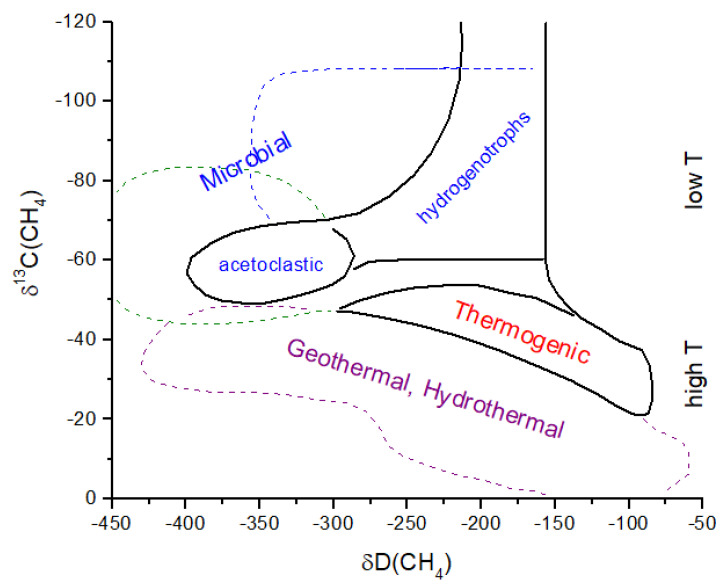
C and H isotope ratios for various biogenic and abiogenic CH_4_ processes. The fields defined here for microbial, thermogenic, and geothermal CH_4_ are from Whiticar (2020). I assume that the earliest Earth (4.4 to 4.0 Gyr) had thermogenic and/or geothermal CH_4_ sources and that hydrogenotrophic methanogens were present after 4.0 Gyr. Acetoclastic and methylotrophic methanogens were also important sources of CH_4_ prior to the GOE. In general, abiogenic processes occur at higher temperatures than biogenic processes.

**Figure 12 life-15-00398-f012:**
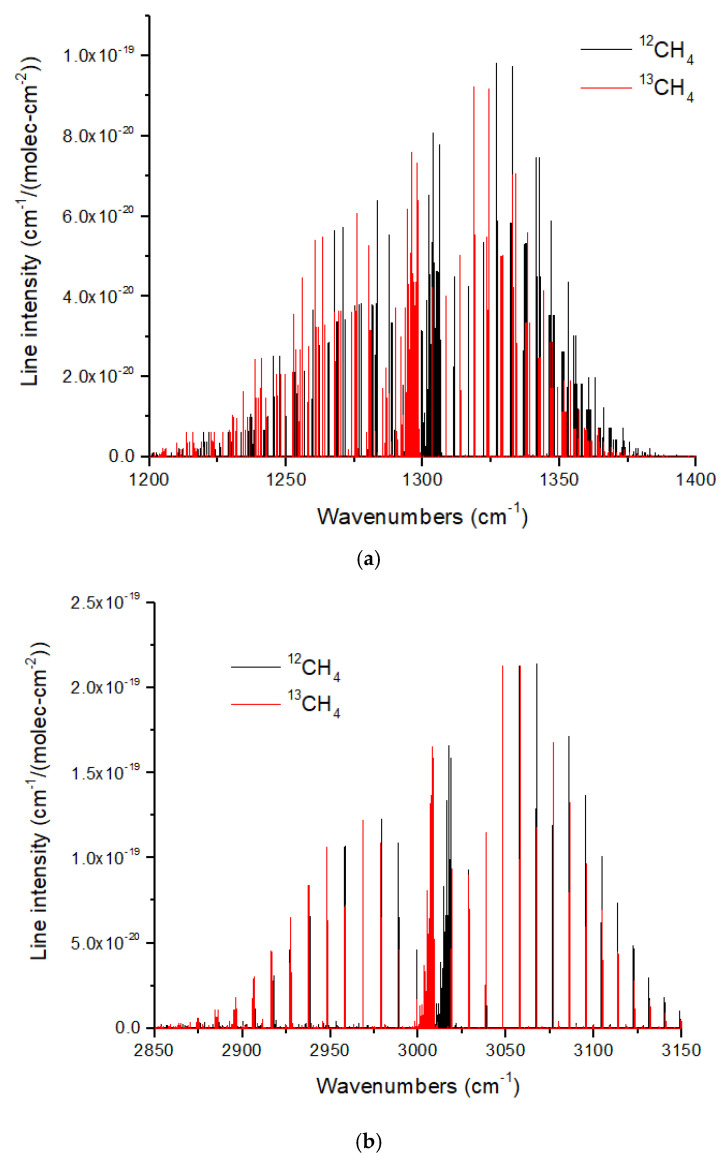
(**a**) Normalized line intensities of ^12^CH_4_ and ^13^CH_4_ in the ν_4_ band, and (**b**) in the **ν**_3_ band. The ν_3_ band has a redshift of ~10 cm^−1^ for ^13^CH_4_, and the ν_4_ band has a redshift of ~8 cm^−1^. For both these figures, the ^13^CH_4_ line intensity has been normalized by the ^13^C fraction (0.011103) given in HITRAN. Data from HITRANonline [66].

**Figure 13 life-15-00398-f013:**
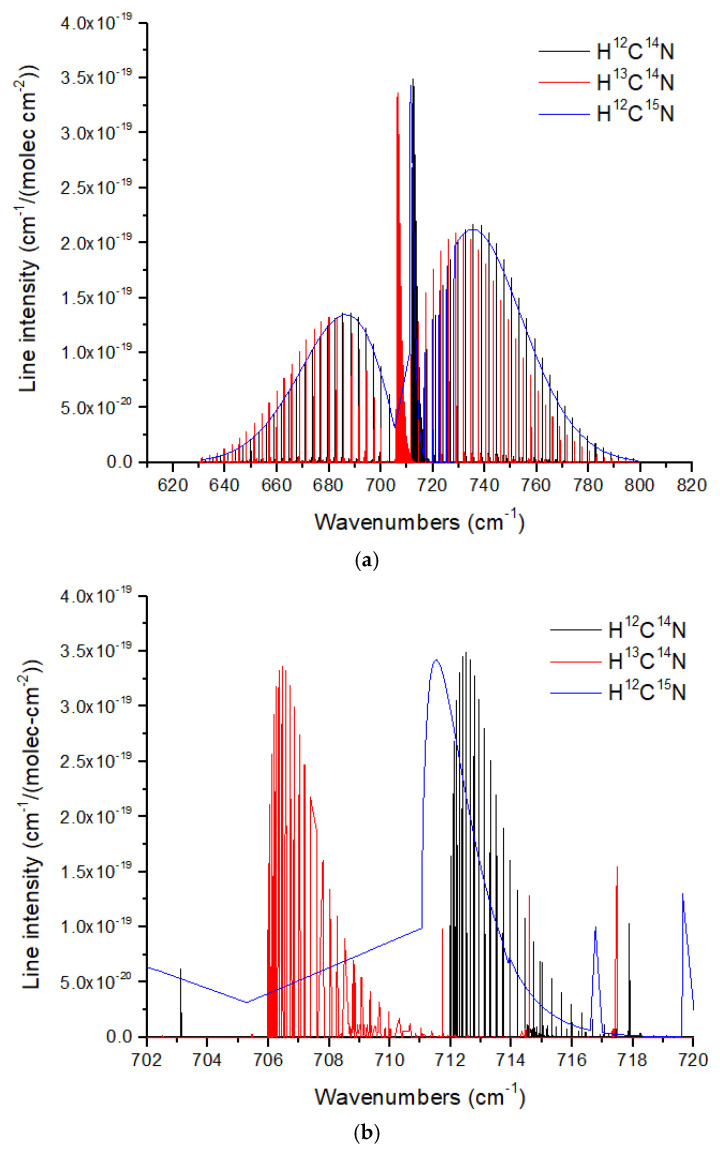
(**a**) Normalized line intensities for the ν_2_ band (bending mode) of three isotopologues of HCN. (**b**) A zoomed-in view of the Q branches illustrating the very small band shift for HC^15^N. Note that the HITRAN data for HC^15^N are incomplete and show only the envelope of the Q-branch. For both figures, the H^13^CN and HC^15^N line intensities are normalized by the ^13^C fraction (0.011068) and ^15^N fraction (0.003622) given in HITRAN. Data from HITRANonline [66].

**Figure 14 life-15-00398-f014:**
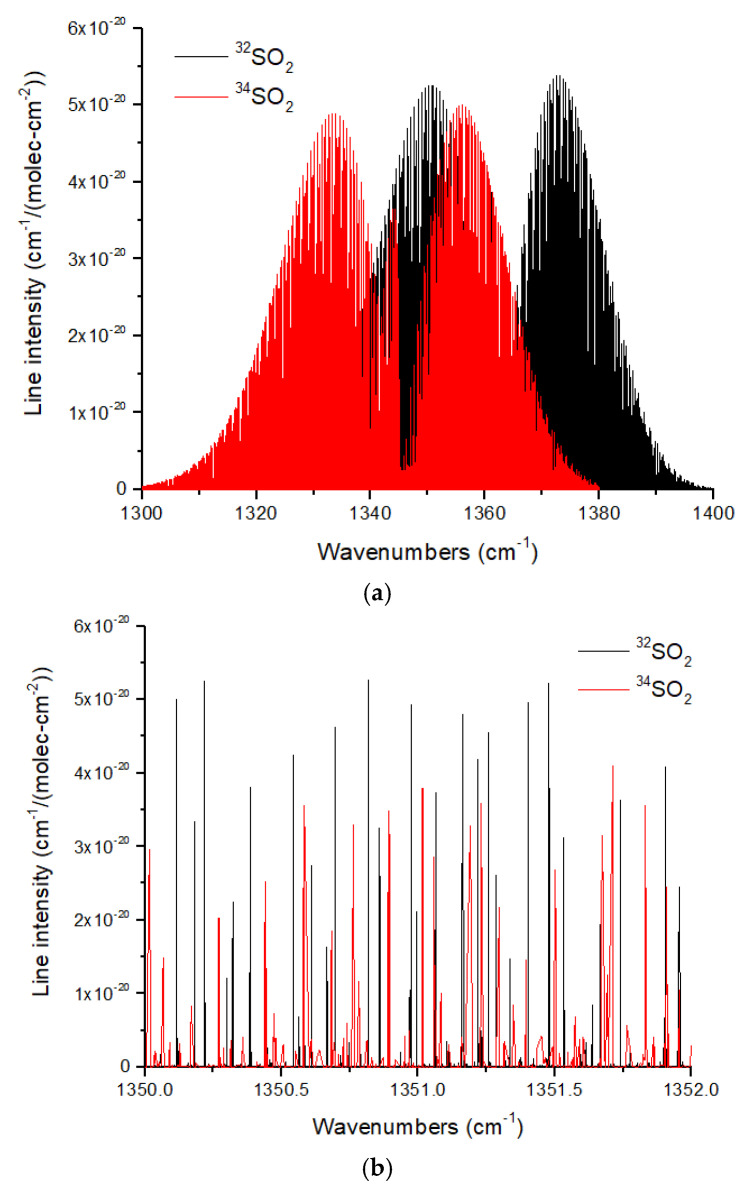
(**a**) Normalized line intensities for the ν_3_ band (asymmetric stretch) of ^32^SO_2_ and ^34^SO_2_. (**b**) Line intensities illustrating the high line density typical of SO_2_ over a range of just 2 cm^−1^. For both figures, the ^32^SO_2_ and ^34^SO_2_ line intensities are normalized by the ^32^S fraction (0.945678) and ^34^S fraction (0.041950) given in HITRAN. Data from HITRANonline [66].

**Table 1 life-15-00398-t001:** Summary of possible photochemical and biological isotopic signatures of exoplanets.

Exoplanet Type	Isotope Ratio	Fractionation Process	Molecule (s)	δ-Values ^a,b^ (‰)	IR Bands	Spectral Resolution ^e^
Exo-Venus	^13^C/^12^C	CO_2_ + hν CO self-shielding	CO mesosphere CO mesosphere	δ^13^C ~−200 probably small	CO 4.7 μm ^f^ CO 4.7 μm	3000–5000 3000–5000
Exo-Earth	^15^N/^14^N ^13^C/^12^C ^13^C/^12^C	N_2_ self-shielding H. methanogens CO_2_ + hν	HCN or HNC CH_4_ vs. CO_2_ CO mesosphere	δ^15^N ~370–5400 Δδ^13^C ~−95 δ^13^C ~−200	HCN ν_3_ see text CO 4.7 μm	700–7000 400–60,000 3000–5000
Super-Earth (rock)	^13^C/^12^C ^15^N/^14^N ^13^C/^12^C	CO_2_ + hν CO self-shielding N_2_ self-shielding H. methanogens ^c^	CO mesosphere CO mesosphere HCN or HNC CH_4_ vs. CO_2_	δ^13^C ~−200 probably small δ^15^N ~370–5400 Δδ^13^C ~−95	CO 4.7 μm CO 4.7 μm HCN ν_3_ see text	3000–5000 3000–5000 700–7000 400–60,000
Sub-Neptune Warm Neptune ^d^	^13^C/^12^C ^15^N/^14^N	CO_2_ + hν CO self-shielding N_2_ self-shielding (low NH_3_)	CO stratosphere CO stratosphere HCN or HNC	δ^13^C ~−200 probably small δ^15^N < 370–5400 at high temps	CO 4.7 μm CO 4.7 μm HCN ν_3_	3000–5000 3000–5000 700–7000
Hot Jupiter	^34^S/^32^S ^13^C/^12^C	SO_2_ self-shielding CO self-shielding	SO_2_ stratosphere SO stratosphere CO stratosphere	δ^34^S ~−10 to −50 δ^34^S ~100 to 200 probably small	SO_2_ ν_3_ SO CO 4.7 μm	6700–47,000 >1000 3000–5000

^a^ For photochemical reactions, δ-values are relative to the parent molecule (e.g., CO_2_, N_2_, SO_2_). ^b^ For hydrogenotrophic methanogens Δδ^13^C = δ^13^C(CH_4_) − δ^13^C(CO_2_). ^c^ For low-temperature super-Earth (~300 K). ^d^ Assuming that hazes do not substantially block UV. ^e^ When the spectra have Q branches, low values are for Q-branch envelopes and high values are for individual Q-branch lines. ^f^ The first overtone at 2.35 µm is also useful.

## Data Availability

The original contributions presented in this study are included in the article. Further inquiries can be directed to the corresponding author.

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
