# Peer review of "Biological, Equilibrium and Photochemical Signatures of C, N and S Isotopes in the Early Earth and Exoplanet Atmospheres"

_life, 2025, doi:10.3390/life15030398_

Round 1
Reviewer 1 Report
Comments and Suggestions for Authors
This paper attempts to model the stable isotope composition of CH4 and other molecules in Earth's atmosphere, with the goal of using spectroscopic observations of exoplanet atmospheres to detect signs of life. The novel contribution is its presentation of estimated carbon stable isotope compositions for atmospheric CO2, CH4, and CO over Earth's 4-billion-year history (Fig. 9).
I recommend revising the introduction before publication, as the current introduction section lacks a clear statement of research objectives. In particular, the role the three sections (biological, equilibrium, and photochemical processes) should be clearly stated in the end of introduction section.
At this version of the manuscript, it is unclear for readers how their findings are reflected in the final conclusions. If Fig. 9 represents the final conclusion, its most distinctive outcome is that CH4 isotope ratios vary across different geological periods. This point should be emphasized, and the reasonings should be explained in more detail. One of the concerns is the author has not considered the large isotope fractionation caused by biological methane oxidation, which may possibly change the conclusion (i.e. Fig. 9). I recommend addressing this point before publication.
While the paper seems to adequately cover previous research, proper citations for necessary fractionation factors are missing. There are many places where key previous studies should have been cited.
Minor points:
L 43. “One of my objectives ….”
What are the others? Objectives should be stated comprehensively here.
L 63-64.
“f” and “j” not appears in the equation (1). Please correct. Does the “C” denote C in the equation?? Also, “d” should be symbol font throughout the text.
L 74.
What is the fate of CH3 radical produced by the reaction (2)? If CH3 react back into CH4 and not converted into CO2, then the overall model may not predict accurate CH4 concentration. Please add rational explanation of the model.
L 113.
Where does the fractionation factor (1.0039) come from? Please add proper citation here. Also, biological CH4 oxidation should be considered because of its larger isotopic fractionation. cf. Gupta et al. 1997: doi.org/10.1029/97GL02858
L 196, 200, 206 and others.
What is Dd13C ? The “D” should be symbol font??
L 239. “Self-shielding of N2 and CO”
Is the atmopspheric total pressure thin enough to cause the isotopic self-shielding? How has the pressure broadening of the N2 and CO treated in the model? This point is not clear to me. Also, the papers cited at L 249 seem dealing with the shielding effect somewhat differently from that written in this manuscript.
L 296
“j-14” should be j-28 as written in equation 23. “j-15” as well.
L 348
Also, laboratory experiment of Ueno et al. (2024: doi.org/10.1038/s41561-024-01443-z) demonstrated the strong 13C depletion of CO and concluded that the actual fractionation is about 30‰ smaller than those calculated from theoretical spectra of Schmidt et al. (2006).
L 365
Need appropriate citation for this notion. This reaction causes about 10‰ fractionation (Feilberg et al., 2005: doi.org/10.1039/B503350K). However, oxidation through termolecular reaction like CO + O + M causes reverse isotope effect, rather resulting more 13C-depleted CO (Ueno et al., 2024).
L 487
Why is it proper to assume the d13C-DIC should be the same as modern value? The reasoning is not clear.
L 520 “Schematic”?
The figure 9 looks important outcome of this paper, though it is not clear which part of the curve is the model result and not assumption? Please explain your results more carefully and explicitly. Especially, why the d13C-CH4 increase to the modern value at around GOE in this model?
From L 534.
I have not reviewed the entire section 6 (Spectral Signatures of Isotopic Gases) in detail. Hope to have another specialist review for exoplanet spectroscopy.
Author Response
I thank the reviewer for their many helpful comments. I address them in order below, where I have assigned numbers to them according to how they are listed in the review report.
- ‘Revise the Introduction’ - Agreed. I have revised the Introduction to more clearly describe the purpose of the paper and how it is organized.
- ‘Final conclusions’ - The Conclusions have been rewritten to more clearly reflect the results of the paper. The overall purpose of this paper is to assess isotope fractionation in exoplanet atmospheres as a potential biosignature. Since rocky exoplanets are of interest, it made sense to start with early Earth. That’s when I realized that not much has been published on early Earth atmospheric isotope ratios. This was not meant to be the primary concluding result of the paper, but it is an interesting one.
- ‘Biological methane oxidation’ - Yes, this is a good point. I’ve added a few sentences discussing the possibility of anaerobic oxidation of methane by methanotrophs.
- ‘Citations for fractionation factors’ - I’ve included a citation for a fractionation factor for CH4due to anaerobic methane oxidation. I’ve clarified the origin of another fractionation factor the reviewer asked about below. Other fractionation factors are either mentioned or implied from Figure 8 (e.g., for methanogenesis).
- L43 - ‘Objectives’ I’ve added text ti the introduction to clarify the objectives of this manuscript.
- L63-64 - ‘equation 1’ As I explained for Reviewer 2, I submitted the original manuscript as a Word doc but not using the Life template. Life editors very kindly transferred the manuscript to the template, but not all fonts transferred correctly. I’ve fixed those typos, including those for equation 1.
- L74 - ‘CH3 radical’ The fate of the CH3 radical for equation 2 in the modern atmosphere is to form H2CO, CO and finally CO2. It will not reform CH4. Even in the ancient atmosphere, CH3will convert primarily to H2CO.
- L113 - ‘fractionation factor 1.0039’ - This value is from Saueressig et al. 2001. I’ve made this clear in the text. I’ve also added the Gupta et a 1997 reference.
- L196,200, 206 - Dd13C has been written in the correct symbol font throughout the manuscript.
- L239 ‘self-shielding of N2 and CO’ - N2self-shielding occurs for Earth at pressures less than 1 microbar (see Figure 3). Pressure broadening is completely negligible at such low pressures. Doppler broadening is far more significant, but not enough to alter the self-shielding effect. The second part of the question asks how self-shielding is treated by Liang et al. (2007) and Yoshida et al. (2023). Yoshida et al. Are evaluating C isotope fractionation due to photolysis of CO2. Because CO2 has a very broad spectrum, they simply used the cross sections for 12CO2 and 13CO2 Liang et al. (2007) used N2 cross sections with a .003 nm resolution. Shielding functions are an alternative to both of these methods, but these are details I’m not going to describe in the paper.
- L296 ‘j-14’ - Again, this was a font change when the manuscript was put into the template. The ‘j’ is actually a jfor flux (note the units). These have been corrected.
- L348 ‘Ueno et al 2024’ - Ueno et al. 2024 has been cited right after equation 29.
- L365 ‘Need citation’ - I’ve added the Feilberg 2005 reference. CO + OH causes a small fractionation, but what I was referring to in the original text was that the reaction returns CO back to CO2, which acts to erase the original photolysis-derived fractionation from CO2. O + CO + M is a slow spin-forbidden 3-body reaction. I agree that it has an interesting isotope effect, but I don’t believe the reaction is important compared to CO + OH in the 50-80 km region of Earth’s atmosphere.
- L487 ‘d13C of DIC’ - Yes, I did not present the argument for why d13C of DIC is not much changed from today. The factors affecting the C isotope composition of DIC, especially in the ocean surface, include 1) the d13C value of atmospheric CO2, 2) the exchange of CO2across the air-sea interface, and 3) photosynthesis. Air-sea exchange is the primary factor that causes atmospheric CO2 to be depleted in 13C compared to DIC. This has been added to the text.
- L520 - ‘schematic’ - Figure 9 is the result of a semi-quantitative model that is based on the arguments presented in the text. A more quantitative model could include a detailed treatment of C isotope fractionation due to air-sea exchange at various temperatures and pH values; the effects of anaerobic and aerobic oxidation of methane; a more careful assessment of photosynthesis. The specific question about why d13C rises near the GOE is because I’ve assumed acetogenic methanogenesis becomes dominant over hydrogenotrophic methanogenesis at this time. Clearly other histories are possible.
- L534 - section 6 - I think one of the other reviewers addressed this section.
Reviewer 2 Report
Comments and Suggestions for Authors
In this work, the author provides a detailed assessment of the signatures of isotopic fractionation due to various biological and abiotic processes on a variety of worlds. It is an interesting and comprehensive work, but there are several items that should be addressed, as described below, before the manuscript can be reconsidered for publication.
1. Lines 26-27 "Stable isotope ratios...on life on Earth" do not have supporting references for corroboration. It is worth citing some astrobiology textbooks and monographs, since they delve into isotope ratios with regards to the origin & evolution of life, biosignatures, and so on; a few examples are provided below:
https://link.springer.com/book/10.1007/978-3-319-96175-0
https://www.cambridge.org/9781009411219
https://mitpress.mit.edu/9780262047661/worlds-without-end/
2. On pg. 1, the author mentions detectability only in passing and focuses thereafter on the significance of measuring isotope ratios, but the former is the more crucial issue at the present. It is recommended to cite works (see also references in point #1 above) that discuss prospects for detection of atmospheric biosignatures in the next few decades.
https://www.liebertpub.com/doi/10.1089/ast.2017.1733
This point is briefly addressed in the Conclusion, but is also worth touching on in the Introduction.
3. There have been several papers on sub-Neptunes (which are habitable) and detecting isotopologues therein. The detection is more feasible for these worlds, because of their larger size and chemical inventories. Please search for and cite such references (e.g., from S. Seager's group).
4. Author should write about why "respiration does not" contribute to isotopic fractionation (line 58).
5. Equation (1) is confusing. It contains delta-13C and phi, but below this equation, there is reference to "j" and "f", neither of which appear in the formula. It feels like material from two different sources has been spliced together. This major issue recurs for other equations too. Please carefully redo all formulas, and ensure that they are correct, and that all variables are properly defined and explained.
6. As noted in point #5 above, there are undefined variables in equation (9). What is "VPDB"[Vienna Peedee Belemnite] for a general reader? Likewise, it is not clear to the reader how equation (9) is obtained, whereas equation (8) can be inferred/understood from the text.
7. In line 136, fractionation factor is called "a" and later it is called "alpha". Likewise, "e" and "epsilon" are used interchangeably" in lines 145-146; and later on, "b" and "beta". This kind of confusing notation should be eliminated - the Greek and Latin alphabet are not equivalent.
8. Equations (10) to (18) are not sufficiently elaborated, and there are several typos. Please include more clarifying text to make this section more self-contained. What is, for instance, m' in equation (17)?
9. Looks like Figures 1a & 1b depict Arrhenius behavior. It may be useful to spell it out, if correct.
10. I understand why the author introduced the notation "Dd13C", but it feels quite confusing from the perspective of an external reader - due to the presence of two Ds here. Perhaps one of them could be replaced with the uppercase Greek "Delta".
11. In line 195, the author states "Cooler, terrestrial exoplanets are likely to have..." but without justification in the form of corroborating references. Please add them.
12. In the discussion of self-shielding (Sec. 4), an interesting point is that it has been extensively discussed in connection with protoplanetary disks (lines 248-249).
13. Equations (19) and (20) contains Theta functions, whereas the text below that makes a reference to Qs. This type of inconsistency has already been highlighted, and must be addressed. Moreover, the rationale/meaning of these equations must be briefly described.
14. Author has focused only on photochemistry, but the ancient solar wind was quite extreme (see reference below) and thus facilitated many different types of atmospheric chemistry reactions and phenomena. Ion escape was prominent in this period. Although not tackled here, it is worth noting at some point and adding suitable references.
https://iopscience.iop.org/article/10.3847/2041-8213/aac489
15. A lot of the photochemistry discussion is similar to the contents of Catling & Kasting (2017). This book can be cited somewhere in the text.
https://www.cambridge.org/core/books/atmospheric-evolution-on-inhabited-and-lifeless-worlds/CB3EE1D3F18A1DB234342E1FF410FC61
16. Why is there a part of the text highlighted in red in line 389? On pg. 12 is the author stating that self-shielding effect - leading to enrichment of heavier isotopes in certain products - destroyed by certain reactions (e.g., exchange reactions) that essentially "reverse" this effect? If so, this point deserves to be reiterated.
17. In referring to d13C values in lines 463-470, is this referring to the C associated with CH4? Just as in Figure 8, it is worth including explicitly.
18. Moreover, in Figure 8, the biogenic values of d13C are, on average, fairly higher than the abiotic sources, though there is some overlap: namely, "≳50" for biogenic vs "≲40" for hydrothermal/thermogenic. It is not clear, therefore, that the "Bulk insoluble organic matter (IOM) in carbonaceous chondrites" would exhibit the same range of d13C as biotic materials. Please also add reference(s) for d13C in carbonaceous chondrites.
19. For HCN in lines 559-568, can the author comment on what kind of spectral resolution would be needed, akin to how it was approximately calculated for CH4 in lines 557-558? It should be easy enough to add. Likewise, it is worth tackling for SO2 as well.
20. Figures 10-13 are quite self-evident, but the corresponding implications should still be discussed a bit more in the text.
21. In closing, I know that much of the discussion is for exoplanets, but it would be interesting to briefly delineate how isotopic fractionation is already a major potential biosignature in the portfolio of "in-situ biosignatures" (see, e.g., the references in point #1).
22. In lines 659-660, the author discusses that "Dd13C" for prebiotic sources is 25‰, whereas it is 95‰ due to methanogenesis. These estimates demonstrate that life is enhancing this value, which may appear to contradict the statement that life would "bring isotope ratios closer to isotopic equilibrium" because some readers will interpret a higher value as corresponding to a greater deviation from equilibrium. I believe that this paragraph could benefit from some rewriting.
Author Response
I thank the reviewer for their many helpful comments, which I address in order below.
- I’ve added some Earth related isotope paper references. I don’t have copies of any of the 3 books mentioned, but I’ve added Biosignatures for Astrobiology as a general reference. I do prefer citing papers when possible.
- Yes, detectability is a topic I would like to discuss in greater detail, but I will defer this to the next paper. I’ve added a couple of sentences and also the excellent reference mentioned by the reviewer.
- I’ve decided to not add separate text on sub-Neptunes. They are indirectly discussed through some of the references (e.g. Glidden et al. 2023).
- I’ve added a few sentences and 1 reference to clarify the effects of photosynthesis and respiration on C isotope fractionation.
- I’ve cleaned up the equations. I originally submitted the manuscript as a Word doc without using the Life template. The journal staff kindly transferred my manuscript to the template, which altered some of the fonts. I’ve corrected those fonts throughout the manuscript.
- Variables and VPDB has been defined for equation 9. I added a brief description of where equation 9 comes from, but without an explicit definition of d13C.
- Same as 5.
- Typos have been corrected. Clarifying sentences have been added at the beginning and end of this sequence of equations, as has a definition for m and m’.
- The temperature dependence of Figures 1 a and b would not be considered Arrhenius behavior because these are equilibrium calculations. Arrhenius behavior occurs due to activation energies in kinetic reaction pathways. In the equilibrium case, this quasi-exponential behavior arises from the vibrational partition function. I’ve added some text to make that point.
- Yes, this was another typo arising from the font change. They are now correctly written as Dd13
- A reference has been added for the expectation of N2 atmospheres for cooler rocky exoplanets.
- Yes, self-shielding was first identified in molecular clouds, and has been applied to protoplanetary disks on numerous occasions. The Lyons and Young (2005) citation is to self-shielding in CO for O isotopes in the solar nebula. The wording of the citing sentence has been modified and a reference has been added.
- The Q’s have been corrected back to thetas. Some explanatory text has been added.
- Yes, I have not considered ion escape processes, including escape associated with collisions due to pick-up ions. Mars is a small and weakly magnetized planet, more vulnerable to solar wind erosion than are larger planets. A sentence has been added to make this clear. I’ve also added the Dong et al reference. I’m interested in Mars O isotope evolution (but not in this paper), so I fully appreciate the 3-D modeling in Dong et al.
- I don’t have a copy of the Catling and Kasting textbook, and I’ve never read it. I’m sure it’s an excellent book. My PhD work was with Yuk Yung in planetary photochemistry, so I know the topic. The photochemistry of N2atmospheres (Earth) was worked out in the 1960’s and 70’s. I prefer to cite papers as needed, rather than textbooks, but if I were to cite a textbook, it would be to Chamberlain and Hunten or one of the other older texts.
- I don’t know how the red highlight appeared, but it has been removed.
- Yes, this is d13C for atmospheric methane. This has been made clear in the text.
- I’ve modified this sentence to include both IOM from carbonaceous chondrites and ordinary chondrites and their d13C ranges. I’ve modified the text to account for these changes. A reference to Alexander et al. 2007 has been added.
- Yes, good suggestion. I’ve added resolution requirements for HCN and expanded the CH4 discussion to include both the Q branch envelope and the individual lines. So2 has also been added.
- In answering 19, I’ve expanded the discussion of the spectral plots.
- In-situ biosignatures is an enormous topic for early Earth. I’ve mentioned it a bit with respect to C isotopes, but it’s too big to get into here. I’ll add a sentence to the Intro that makes this clear.
- The key to this statement is that life may help a system reach isotopic equilibrium at LOW temperatures where equilibrium is generally not possible. Is life acting as an isotopic catalyst?
Reviewer 3 Report
Comments and Suggestions for Authors
The topic of the present research is appealing. The final design does not benefit references to major works which have contributed in understanding the isotopic composition of planet Earth and the changes of isotopic compositions of different reservoirs from 4.6 Ga to present. In the PDF file some comments are insert. For example the presentation of the carbon isotopic composition of CO2 and its evolution in time is poor treated.
Conclusions are also containing new references and sounds rather like a resume of previous works than to present the findings of the present research. Should be rewritten concentrating on the new findings of the present investigation

Author Response
I thank the reviewer for their comments, addressed in order below.
- Unfortunately I did not have a chance to check the Sharp 2018 reference.
- Typos corrected. They occurred when the manuscript was transferred to the template.
- Yes, anthropogenic CO2 is important for d13C of CO2 in modern atmosphere.
- Reference added.
- Repetition removed.
- Des Marais 2001 (RiMG chapter) has been added as a reference. It’s an excellent chapter. I’m not sure that I understood this comment. In the Archean atmosphere d13C of CO2 is set by volcanic output. Equilibration of CO2 gas with surface water results in DIC that is ~ 8 permil or so more positive than the CO2 gas. Apart from temperature and maybe small pH changes, would this description of d13C of CO2 gas and DIC not remain the same through the Archean? Perhaps I’m missing an important point here.
- Conclusions and Discussion have been separated.
Reviewer 4 Report
Comments and Suggestions for Authors
The manuscript
Biological, Equilibrium and Photochemical Signatures of C, N
and S Isotopes in the Early Earth and Exoplanet Atmospheres
represents
a well-written and relevant article. It is relevant for the broader Astrobiology, exoplanetary science and planet formation communities. I recommend publication after refining it according to my comments below.
COMMENTS
Line 272
in prep not relevant. Please remove
Figure 3
What is SZA? Please define, and also J again in the figure caption.
Line 289
Red box
Line 624
insufficient photons -> insufficient number of photons?
Author Response
I thank the reviewer for their comments, which are addressed in order.
- The mention of the in prep work is relevant to the student that I’m working with on that topic. I would like to keep it if possible.
- SZA (solar zenith angle) has been defined in the relevant figure caption.
- Red box has been removed. I’m not sure where that came from.
- Insufficient number of photons has been corrected.
Round 2
Reviewer 2 Report
Comments and Suggestions for Authors
The author has addressed most (although not every single out) of my queries and suggestions for improvement satisfactorily. I recommend publication of the manuscript in its current form.
Author Response
My kind thanks to the reviewer.
Reviewer 3 Report
Comments and Suggestions for Authors
Methods or Results chapters are missing. The contribution looks like a Resumee of previously published works on the topic.
Author Response
reviewer: Methods or Results chapters are missing. The contribution looks like a Resumee of previously published works on the topic.
response: Methods and results are presented in each of sections 2 though 6 because I am addressing the topic of isotopes in exoplanet atmospheres from multiple perspectives. There have been very few papers published on this topic, and certainly nothing this comprehensive.